# Regulation of ADP-ribosyltransferase activity by ART domain dimerization in PARP15

Carmen Ebenwaldner [1], Antonio Ginés García Saura[1], Simon Ekström [2], Katja Bernfur [1], Martin Moche[3], Derek T. Logan [1], Michael S. Cohen [4] & Herwig Schüler [1] ✉

PARP15 is a mono-ADP-ribosyltransferase that targets an unknown set of proteins as well as RNA. Its evolutionary relationship with PARP14 suggests roles in antiviral defence; its localization to stress granules points to functions in the regulation of translation. Here we show that the transferase domain of PARP15 dimerizes in solution; the formation of dimers is a prerequisite for catalytic activity and monomeric mutant variants of the domain are inactive. In cells, dimer-disrupting mutations abrogate catalytic activity and alter the subcellular localization of the full-length protein. Using biophysical methods, including X-ray crystallography and HDX-MS, we provide evidence for a regulatory mechanism by which dimerization enables correct target engagement rather than NAD$^+$ co-substrate binding, and by which the two protomers of the dimer operate independently of one another. Together, our results uncover a regulatory mechanism in a PARP family enzyme.

Proteins of the poly(ADP-ribosyl) polymerase (PARP) family are defined by a common, structurally highly conserved catalytic ADP-ribosyltransferase (ART) domain[1]. ADP-ribosylation, in which the ADP-ribose moiety from NAD$^+$ is transferred onto target molecules, participates in the regulation of processes including DNA damage repair, transcription and translation, protein homeostasis, and antiviral defense[1–4]. ADP-ribosylation is of prime therapeutic interest due to its contribution to DNA damage repair and de-regulated signaling pathways in cancers[5,6]. Mechanisms that regulate PARP enzymatic activities are bound to be different for every PARP family member, as these differ in the types and numbers of accessory domains. Such domains include recognition modules for DNA, RNA, and ADP-ribosylated macromolecules, as well as protein interaction modules[7].

The subfamily of macrodomain-containing PARP enzymes (PARP9, PARP14, and PARP15) was originally identified as B-aggressive lymphoma (BAL) proteins 1-3[8]. PARP9 and PARP14 contain a macrodomain with ADP-ribosyl hydrolase activity[9–11] as well as one and two ADP-ribosyl binding macrodomains, respectively[12,13]. PARP15, the third member of the subfamily, originated from partial duplication of the structurally more complex PARP14 early during mammalian evolution[14]. Both genes have remained under positive selection pressure, but the *parp15* gene was lost again independently in some lineages, and truncated or duplicated in others. Thus, evolutionary analysis strongly ties PARP15 to virus defense[14]. Humans have two isoforms of PARP15; isoform-1 contains an unstructured N-terminal extension of unknown function, two macrodomains, and a C-terminal ART domain, whereas isoform-2 features a shorter N-terminal extension and lacks macrodomain-1 but is otherwise identical to isoform-1. Macrodomain-1 is poorly characterized, whereas crystal structures and biochemical analyses suggest that macrodomain-2 is an ADP-ribosyl binding module[12,13]. PARP15 has been shown to localize to stress granules[15], cytoplasmic condensates consisting of proteins and RNA that form upon exposure to a variety of stressors, including viral infection[16]. More recently, PARP15 was found to ADP-ribosylate 5'-phosphorylated RNA ends in vitro and in cells[17,18].

Despite the recent evolutionary kinship of PARP14 and PARP15, the PARP15 ART domain has a higher affinity for NAD$^+$ and modifies more sites on itself in vitro[19,20]. Here, we set out to study the molecular basis for

[1]Division of Biochemistry and Structural Biology, Department of Chemistry, Lund University, Lund, Sweden. [2]SciLifeLab and BioMS, Integrated Structural Biology platform, Structural Proteomics Unit Sweden, Lund University, Lund, Sweden. [3]Protein Science Facility, Department of Medical Biochemistry and Biophysics, Karolinska Institutet, Stockholm, Sweden. [4]Department of Chemical Physiology and Biochemistry, Oregon Health and Science University, Portland, OR, USA. ✉e-mail: herwig.schuler@biochemistry.lu.se

these differences. We show that the PARP15 ART domain forms a dimer in solution and that dimer formation is required for mono-ADP-ribosylation (MARylation) activity. We show that the dimer interface in solution is identical to the interface that has been observed by X-ray crystallography. Mutation of a central salt bridge network formed by R576 and D665 prevents dimerization, abolishing the catalytic activity of PARP15 in vitro and in cells. We find that the activation by dimerization of one protomer is not dependent on a functional catalytic site in the second protomer. We do not see any evidence for NAD[+] binding cooperativity in the PARP15 dimer being the cause of catalytic activation, but a crystal structure of the PARP15 ART domain mutant R576E shows differences compared to the wild type in the D-loop that covers the active site. Finally, by HDX-MS analysis, we find prominent differences in the flexibility of the acceptor sites of the monomeric and dimeric PARP15 variants, which could explain how the dimerization event affects activity. In summary, this study proposes catalytic domain homodimerization as a regulatory mechanism for a member of the PARP enzyme family.

## Results

### PARP15 dimerizes via its ART domain

During routine purification of the human PARP15 ART domain (Asn[481]-Ala[678]), we observed that its elution profile in size exclusion chromatography was inconsistent with a monomer. To characterize this behavior, we carried out size exclusion chromatography in combination with right-angle and low-angle light scattering measurements (Fig. 1a–f, Supplementary Table 1). The results indicated that the PARP15 ART domain eluted as a dimer (Fig. 1a). For comparison, we characterized several protein domains related to PARP15. The PARP14 ART domain (His[1608]-Lys[1801]; 65% sequence identity) was monomeric, and the PARP10 ART domain (N[819]-V[1007]; 37% sequence identity) was largely monomeric (Fig. 1b, c). An amino-terminally truncated PARP15 isoform-2 consisting of macrodomain-2 and ART domain (m2-ART; Gly[287]-Ala[678] of the canonical isoform) eluted in a broad peak that contained both monomeric and dimeric species (Fig. 1d), while macrodomain-2 alone (m2; Gly[287]-Asn[470]) eluted as a monomer (Fig. 1e). To substantiate the interpretation that the PARP15 ART domain was a dimer in solution, we treated proteins with BS3, a chemical cross-linker with a spacer arm length of 11.4 Å[21] and reactivity towards amines (lysine and N-terminus) and hydroxyls (serine, threonine, tyrosine)[22,23]. Using only a half-molar equivalent concentration of crosslinker per protein, PARP15 ART and m2-ART could be crosslinked as dimers, whereas PARP15 m2 and PARP14 ART could not (Fig. 1g).

Single particle sizing experiments using mass photometry suggested that ART and m2-ART constructs were monomeric at low nanomolar concentrations, and the size distributions shifted toward dimers with increasing protein concentrations in a range up to 120 nM (Fig. 1h, i). This supported the interpretation that the PARP15 ART domain is a dimer in solution. Finally, van Holde-Weischet analysis of a sedimentation velocity analytical ultracentrifugation (SV-AUC) experiment performed at four concentrations of PARP15 ART domain reveal a pattern characteristic for a reversible self-associating system[24], with sedimentation coefficients increasing as protein concentration increases (Fig. 1j). To maximize the separation of monomers and dimers, another SV-AUC experiment was performed at 58,000 rpm and at the lowest reliably measurable concentration of the PARP15 ART domain as determined in panel j (0.5 μM; measurement at 220 nm). A Discrete Model Genetic Algorithm (DMGA) analysis[25] of these data suggested a dissociation constant of 313 nM (Fig. 1k, Supplementary Fig. 1, and Supplementary Table 2).

### PARP15 in solution features a dimer interface known from crystallography

Having established that the PARP15 ART domain is a dimer in solution, we were interested in defining the location of the dimer interface.

Abundant crystallographic data is available for this domain in the Protein Data Bank, and with no exception, the domain has crystallized featuring a homodimer in the asymmetric unit[26–29]. Our analysis of 39 structural models using the PISA server[30] gave a mean buried surface area of $1046 \pm 23$ Å[2] for the crystallographic dimer interface (representative example given in Supplementary Table 3). For comparison, a crystal structure of the homodimeric 14-3-3β (PDB: 2BQ0)[31] features a buried surface area between the protomers of 1030 Å[2]. Thus, it was reasonable to assume that the dimer interface present in the PARP15 ART domain crystals may be relevant to the dimer in solution. In all PARP15 ART domain models based on crystallographic data, the dimer interface was made up of side chains in helix α2 of each monomer, as well as a prominent salt bridge network formed by residues in the two turns following helix α3 and between strands β8 and β9, respectively, from each monomer (Fig. 2a-c, Supplementary Table 3). In addition, hydrogen bonds and hydrophobic interactions were formed across the interface by side chains in the vicinity of the sites above. AlphaFold3[32] predicted a dimer with high confidence (ipTM = 0.89, pTM = 0.92), with an interface identical to the one in the crystallographic dimers (Fig. 2a and Supplementary Fig. 2a, b).

To test whether the crystallographic dimer interface was present in the dimer in solution, we analyzed the m2-ART construct using hydrogen-deuterium exchange coupled with mass spectrometry (HDX-MS). The results showed that a surface that coincides with the crystallographic dimer interface is protected from hydrogen exchange during the experiment (2.5 hours; Fig. 2d and Supplementary Fig. 2c,d). Furthermore, we crosslinked the PARP15 ART domain under the conditions of Fig. 1g and analyzed the product by mass spectrometry after proteolytic cleavage. This analysis revealed four high-confidence crosslinks, one of which unambiguously represents an inter-protomer crosslink as it was identified on peptides with overlapping sequences – a crosslink that must have originated from two individual protomers (XL-2 in Supplementary Fig. 3, Supplementary Table 4). All other crosslinks could potentially be intra-molecular (within the same protomer) or inter-molecular (between one protomer and the other). As intra-molecular crosslinks are inherently more prominent in crosslinking experiments, we assume that this is also the case for the non-overlapping peptides identified in our experiment. When mapped on the structure of the crystallographic dimer, all four cross-links are in agreement with the distance constraints of the BS3 crosslinker (Fig. 2e). From these results, we conclude that the crystallographic dimer interface is highly similar or identical to the dimer interface in solution.

### Catalytic activity in PARP15 requires dimer formation

Oligomerization is a common means of protein regulation. Therefore, it was important to investigate whether PARP15 catalytic activity was related to its oligomerization state. To that end, we created several ART domain variants in which either of the two residues R576 and D665, which participate in the interface salt bridge network (Fig. 2b), had been mutated. We made variants with three different substitutions for R576 and two substitutions for D665 (Fig. 3a). All interface mutants were apparently correctly folded, as shown by the ability of the nicotinamide analog 3-aminobenzamide to stabilize these ART domains in thermal shift assays (Supplementary Fig. 4). Each mutant protein construct showed a higher retention volume than the wild-type construct in analytical size exclusion chromatography (Fig. 3b), indicating that they were monomeric. All mutants eluted in one symmetric peak except for R576A, which may retain weak dimerization (Fig. 3b).

To test if the activity of the monomeric variants was affected, we incubated the dimer interface mutant proteins, alone or in pairwise combinations, with NAD[+] (10% biotin-NAD[+]) and analyzed the reactions by Western blotting, followed by detection of auto-MARylation levels using HRP-streptavidin (Fig. 3c). Alternatively, we analyzed catalytic reactions containing NAD[+] (no biotin-NAD[+]) using a GFP-macrodomain

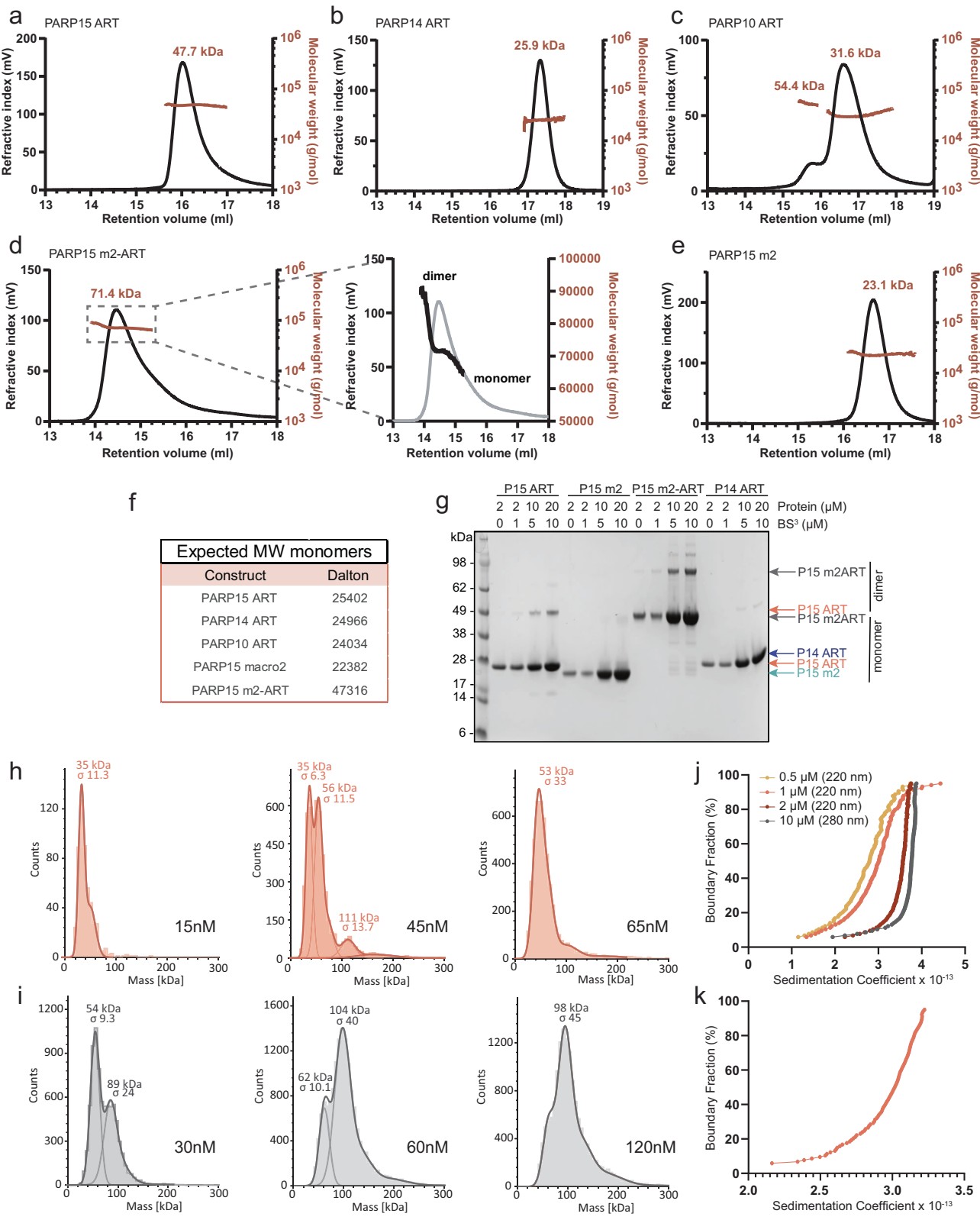

**Nature Communications** | (2025)16:9567

overlay assay (MacroGreen, Fig. 3d)[33]. All mutant variants were less active than the wild-type construct (Fig. 3c, d). These findings suggest that ART domain dimerization is a prerequisite for catalytic activity in PARP15. We verified using a MAR-specific antibody that PARP15 is not auto-MARylated during expression in *E. coli* (Supplementary Fig. 5a). Consistent with our interpretation, the R576A mutant, which displayed residual dimerization (Fig. 3b), also retained the highest level of

activity (Fig. 3c, d). The charge reversion mutants R576D and R576E had no measurable catalytic activity, although thermal stabilities and ligand-induced $T_m$-shifts indicated they were properly folded (Supplementary Fig. 4). Combining equal amounts of the charge reversion mutants at residues R576 and D665 did not result in rescue of MAR-ylation activity (Fig. 3c, d). Combining equal amounts of the charge-to-alanine mutants at residues R576 and D665, however, retained low

**Fig. 1 | PARP15 dimerizes via its ART domain.** SEC-RALS/LALS analysis revealed that the PARP15 ART domain eluted as a dimer (**a**) from the size exclusion column whereas the ART domains of PARP14 (**b**) and PARP10 (**c**) were monomeric. The peak of the PARP15 macrodomain-2 · ART domain construct (m2-ART; **d**) contained both monomeric and dimeric species. The PARP15 macrodomain-2 (**e**) was a monomer. Further details are given in Supplementary Table 1. **f** Expected molecular weights of the respective monomers. **g** PARP15 ART and m2-ART constructs showed dimers on an SDS-PAGE gel after crosslinking with BS3. PARP15 m2 and PARP14 ART did not form dimers. Representative SDS-PAGE of $n = 2$ independent experiments. Size

distribution of single particles of PARP15 ART (**h**; salmon) and PARP15 m2-ART (**i**; grey) in the nanomolar concentration range, observed by mass photometry. **j** Van Holde-Weischet integral distribution plot of sedimentation velocity experiments by analytical ultracentrifugation, conducted at four concentrations of ART domain and 40,000 rpm for 12 h. **k** Van Holde-Weischet integral distribution plot of a sedimentation velocity experiment conducted at 0.5 μM ART domain and 58,000 rpm for 5 h. Analysis of these data suggested a $K_d$ for dimer formation of 313 nM. Source data are provided as a Source Data file.

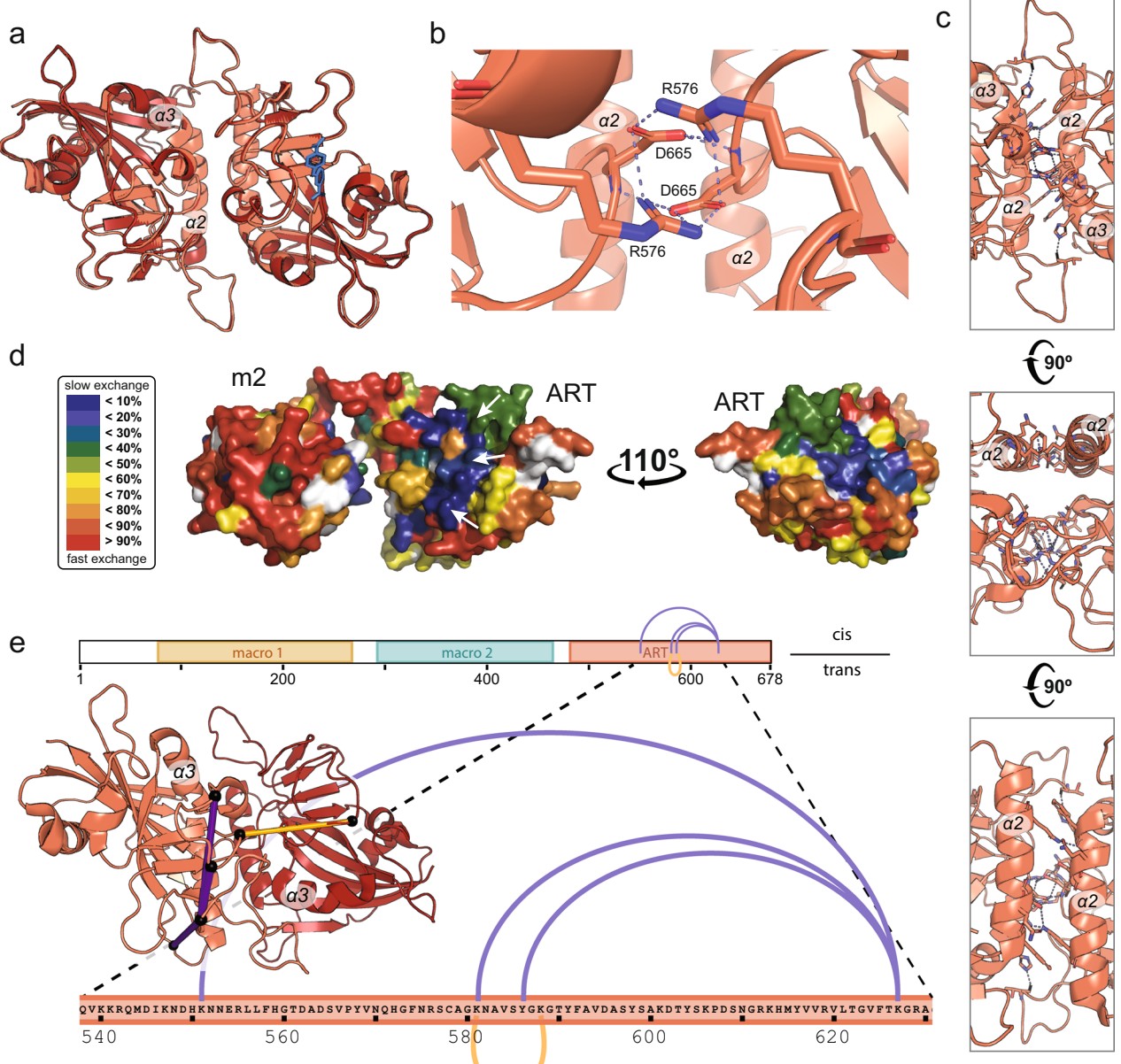

**Fig. 2 | A combination of experimental and computational tools defines the dimer interface. a**–**c** A structural model generated by AlphaFold3 suggests an ART domain dimer (dark red) that superimposes with an $RMSD_{C\alpha}$ of 0.387 Å with the crystal dimer (6RY4[64]; light red). The ligand (blue) is bound to the active site of one protomer in the crystal structure. **b** The dimer interface contains a prominent salt bridge network formed by side chains R576 and D665 of each protomer. **c** Helices α2 of each protomer, as well as residues in the vicinity of α3, interact at the dimer interface. Residues participating in polar interactions are shown in stick representation. **d** AlphaFold3 model of PARP15 m2-ART, colored for protection against

hydrogen exchange at 2.5 h incubation in $D_2O$ (red, fast hydrogen exchange; blue, slow hydrogen exchange). White arrows mark the location of the dimer interface. Detailed illustrations of the HDX-MS data are shown in Supplementary Fig. 2. **e** BS3-crosslinked ART domain peptides identified by mass spectrometry are in agreement with the dimer observed by X-ray crystallography. The inter-protomer crosslink K581-K588 is indicated in yellow; intra-protomer crosslinks K551-K627, K581-K627, and Y586-K627 are shown in purple. Crosslinked peptide sequences and masses are given in Supplementary Table 4. Source data are provided as a Source Data file.

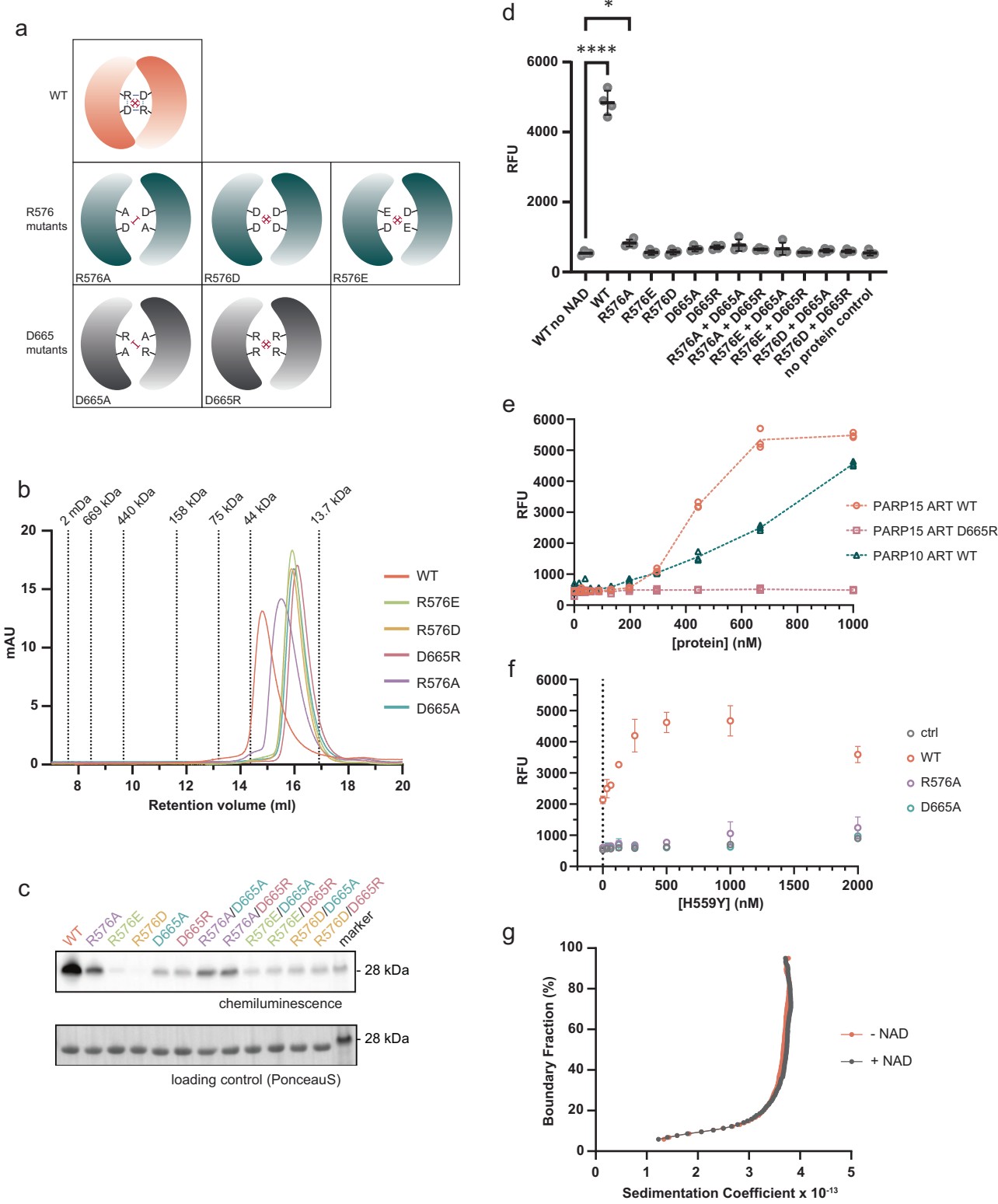

levels of activity, again likely due to weak or transient dimer formation as in the R576A mutant alone. Finally, to test whether trans-modification is impacted by destabilization of the PARP15 dimer, we used SRSF protein kinase 2 (SRPK2) as a target[34,35]. We found that the effect of the dimer destabilizing mutations was similar or even more pronounced in trans-modification compared to auto-modification (Supplementary Fig. 5b).

To further explore the link between PARP15 dimer formation and catalytic activity, we measured auto-modification activity over a range

of enzyme concentrations, for the ART domains of the monomeric PARP10 (Asn[819]-Val[1007]) and of PARP15. Figure 3e shows that PARP10 ART domain auto-modification increased in a roughly linear fashion over the entire concentration range. In contrast, PARP15 ART domain activity displayed a sharp incline at an enzyme concentration above 200 nM. We believe that this behavior reflects the onset of MARylation activity once the enzyme concentration supports dimer formation; this would suggest an apparent dissociation constant in the mid-nanomolar range under these experimental conditions, which is

**Fig. 3 | PARP15 is active as a dimer in vitro. a** Schematic representation of the wild-type dimer with its central salt bridge network consisting of one arginine and one aspartate per protomer. The point mutations introduced are illustrated below, with the expected electrostatic repulsions indicated as red dashes. **b** All interface mutant variants eluted as monomers in analytical size exclusion chromatography. All mutant constructs showed significantly less auto-MARylation activity compared to WT PARP15, as shown by Western blot analysis of biotin-NAD$^+$ spiked reactions (**c**; 10 µM total enzyme) and by a fluorescent macrodomain overlay assay (**d**; 1 µM total enzyme; $n = 4$ replicates, mean ± SD shown; Statistical significance was determined using an ordinary one-way ANOVA (F [13, 42] = 316.0, $p < 0.0001$, R$^2$ = 0.9899), followed by Dunnett's multiple comparisons test comparing each PARP15 variant to the control. Adjusted $p$-values: *$p \leq 0.05$, ****$p \leq 0.0001$ (exact $p$-values are found in the Source Data file)). The combination of variants with different surface mutations could not rescue the loss-of-activity phenotype. **e** Extent of auto-modification in PARP15 and PARP10 ART domain constructs over a range of enzyme concentrations. Whereas PARP10 activity showed a near-linear increase of activity with increasing protein concentration, PARP15 activity increased sharply after 200 nM. $n = 3$ replicates. **f** PARP15 macrodomain-2 was bound to high-binding 96-well plates to serve as target for ADP-ribosylation. Wild-type PARP15 ART, D665A mutant, or R576A mutant, were kept at a constant concentration (250 nM), and increasing concentrations of the catalytically inactive mutant H559Y were added. Control, H559Y mutant only titration (no other PARP15 variant present). ART domains were washed out, and ADP-ribosylation on macrodomain-2 was quantified. The results reveal that the addition of catalytically dead mutant boosts the activity of the WT PARP15 but does not rescue the activity of dimerization-incompetent mutants. Data are represented as mean ± SD; $n = 3$ replicates. **g** Auto-MARylation of PARP15 ART does not influence its oligomeric state, as seen in the van Holde-Weischet integral distribution plot of an SV-AUC experiment performed at 9 µM protein concentration. Source data are provided as a Source Data file.

similar to the dissociation constant of 313 nM determined using AUC under NAD$^+$-free conditions (Fig. 1k). To strengthen this conclusion, we repeated the experiment with the PARP15 dimer interface mutant D665R (Fig. 3e). This monomeric variant remained inactive over the range of concentrations tested.

Having established that the catalytically active species of the PARP15 ART domain is a dimer, we asked whether each protomer within the dimer must be catalytically active for the MARylation reaction cycle to proceed. For that purpose, we coated assay plates with PARP15 macrodomain-2 as a target for MARylation. Either the wild-type ART domain or the monomeric variants D665A and R576A were added at a concentration of 250 nM, close to the apparent dissociation constant of the wild-type dimer. Finally, an ART domain mutant H559Y, which is catalytically inactive[26], was added at increasing concentrations to the wells of the assay plate. The ADP-ribosylation reaction was initiated by the addition of NAD$^+$. Subsequently, excess NAD$^+$ and the wild-type and mutant ART domains were washed off, and MARylation of macrodomain-2 was detected in a fluorescent macrodomain overlay assay (MacroGreen, Supplementary Fig. 5d). Wild-type ART domain alone at 250 nM caused a moderate level of MARylation (Fig. 3f). Despite not contributing to the catalytic reaction, the addition of inactive H559Y mutant to wild-type PARP15 caused a rise in MARylation levels, which reached saturation around 500 nM total enzyme (sum of wild-type and H559Y mutant; Fig. 3f). When wild-type enzyme was substituted by either of the dimerization mutants R576A or D665A, and H559Y mutant was added, no rise in MARylation activity was observed. We conclude that the H559Y mutant construct can participate in heterodimer formation with the wild-type ART domain, thus activating catalysis without processing NAD$^+$. This implies that catalytic activity in each protomer within the dimer is independent of the activity in the second protomer. Importantly, auto-MARylation of the PARP15 ART domain does not alter its oligomeric state, as observed by SV-AUC (Fig. 3g, Supplementary Fig. 5c).

## Full-length PARP15 is active as a dimer in cells

Next, we wanted to test the effect of the dimer interface mutations on PARP15 activity in cells. To that end, we expressed N-terminally mEGFP-tagged full-length PARP15 (isoform 1) in HEK293T cells and performed immunofluorescence to compare MARylation levels produced by the different PARP15 variants (Fig. 4). Four hours post transfection, cells were treated with Saruparib (AZD5305) to diminish PARP1 contribution to ADP-ribosylation levels, and 4 h before fixation, cells were treated with the PARG inhibitor PDD00017273, which enhances PARP15 MARylation in cells. The wild-type protein and the catalytically inactive H559Y mutant established the higher and lower levels of MARylation observed. Cells expressing interface mutants displayed reduced levels of MARylation, with R576A retaining the highest residual activity; the D665 mutants retaining intermediate levels of activity; and the charge reversal mutants R576E and R576D being

comparable to the catalytically inactive H559Y mutant. Also, Western blot assays performed on lysates from transfected cells showed that the MARylation target pattern seen for wild-type PARP15 was absent in catalytically inactive mutants, including the monomeric variants (Supplementary Fig. 6). Thus, overall, the relative MARylation levels observed in PARP15 mutant-expressing cells were as expected based on the MARylation activities of the purified recombinant ART domains in vitro (Fig. 3c,d). Interestingly, the localization pattern of PARP15 appeared to depend on its MARylation activity: While the wild-type protein localized to nuclei and to cytoplasmic foci, the inactive mutants showed a more homogeneous and solely nuclear expression pattern. Mutants retaining marginal MARylation activity showed a granular localization pattern, mostly limited to the nucleus. This suggests that PARP15 localization in the cell may be controlled by its catalytic activity, which is in turn regulated by homodimerization.

## Putative mechanism for activation of catalysis in PARP15 dimers

To understand the mechanism for dimerization-dependent catalytic activation, we reviewed published crystal structures of the PARP15 ART domain. Previous structures showing the wild-type homodimer suggested a possible mechanism by which dimerization might control catalysis: By enabling the salt bridge network at the dimer interface the R576 side chain, situated at the N-terminal base of the D-loop, might lock the D-loop in a catalytically active conformation. To test that hypothesis, we crystallized the monomeric R576E mutant in complex with the nicotinamide analog 3-aminobenzamide (3AB). The mutant crystallized with a symmetry (C 2 2 2$_1$) that had not been observed for the PARP15 ART domain (PDB: 9IFV, Supplementary Table 5, Supplementary Fig. 7). The asymmetric unit contains two chains - each with a 3AB molecule in the NAD$^+$ binding site - that superimpose with an RMSD$_{C\alpha}$ of 0.14 Å but which do not engage as a dimer (Fig. 5a). However, each chain shares an interface with a symmetry mate, revealing chainA-chainA and chainB-chainB dimers in the crystal lattice that share the dimer interface known from the wild-type structures (Fig. 5b). Superimposition with the wild-type PARP15 crystal structure (PDB: 6RY4) shows that the wild-type dimer is highly similar to the mutant (RMSD$_{C\alpha}$: 0.409 Å to the chainA-chainA dimer, and 0.452 Å to the chainB-chainB dimer; Fig. 5c).

In our structural model, the salt-bridge network central to the dimer interface is lacking due to the R576E mutation. Although this did not prevent the protein from crystallizing in a dimeric state, analysis using the PISA server revealed a further de-stabilized dimer interface: The R576E mutation allowed a shift of the D-loop base away from the interface, whereby the distances in C$_\alpha$-atom positions increased by 3.7 Å in S577 and by 3.6 Å in C578. This coincided with loss of multiple hydrogen bonds at the interface (Fig. 5c and Supplementary Table 6). It appears reasonable to assume that these changes together prohibit dimer formation in the mutant protein at physiological concentrations in solution. Given the destabilization of dimer contacts in the R576E

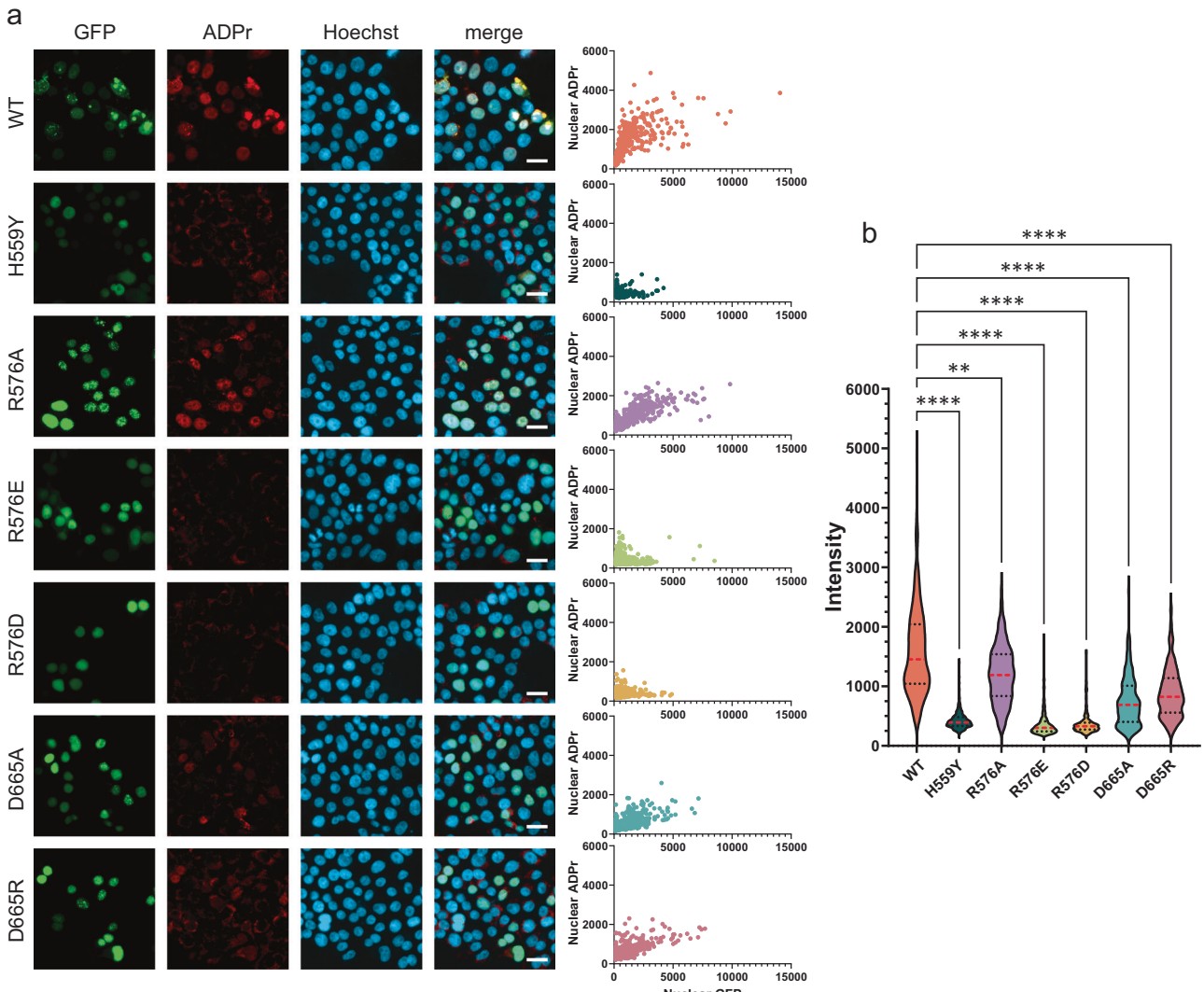

**Fig. 4 | Dimer interface mutations compromise PARP15 activity in cells.** Wild-type or mutant variants of mEGFP-tagged full-length PARP15 were overexpressed in HEK293T cells. 4 h post-transfection, cells were treated with 100 nM PARP1 inhibitor Saruparib and 4 h before fixation with 1 μM PARG inhibitor (PDD 00017273). Cells were stained for ADP-ribosylation (red). **a** Representative micrographs. Scale bar, 20 μm. Scatter plots highlight the correlation of nuclear GFP signal with nuclear ADP-ribosylation levels in individual cells. **b** Quantification of the micrographs in (a), displaying levels of ADP-ribosylation across GFP-positive nuclei. Medians and quartiles are indicated by red dashed and black dotted lines, respectively. Statistical significance was determined using a Kruskal-Wallis test (H[6] = 1213, $p < 0.0001$), followed by Dunn's multiple comparisons test comparing each mutant to WT. Number of cells analyzed: WT ($n = 341$), H559Y ($n = 235$), R576A ($n = 319$), R576E ($n = 409$), R576D ($n = 267$), D665A ($n = 210$), D665R ($n = 198$). Adjusted $p$-values: **$p \leq 0.01$, ****$p \leq 0.0001$. Source data, including exact $p$-values are provided as a Source Data file.

mutant and the displacement of the D-loop base, we hypothesized that PARP15 might employ a NAD$^+$ binding cooperativity across the dimer that was lost in our monomeric R576E variant. To test whether NAD$^+$ binding to one protomer either facilitates or impedes NAD$^+$ binding to the second protomer, we performed isothermal titration calorimetry (ITC) with the wild-type ART domain and the R576E mutant and tested their binding to the general PARP inhibitor EB-47 (Supplementary Fig. 8). Wild-type PARP15 bound to EB-47 with a dissociation constant of 24 μM, while the R576E mutant had a reduced affinity for the ligand, with a $K_d$ of 47 μM. This shows that while the affinity for NAD$^+$ may be decreased, the monomeric variant retained the ability to bind to its co-substrate. Importantly, however, in both wild-type and monomeric mutant, the data showed a ligand occupancy of roughly one mole of EB-47 per mole of protein, suggesting that cooperative NAD$^+$ binding is not part of activity regulation in PARP15.

Crystal structures, while often offering invaluable insights into mechanistic details, are limited to a specific state of the dynamic spectrum of a protein. Thus, we employed HDX-MS to investigate in-solution differences in conformation and flexibility between wild-type PARP15 and its R576E variant upon binding of an active-site ligand. For this analysis we again chose EB-47, a compound that recapitulates the sub-structure of NAD$^+$ and for which experimental structural models are available. The mass spectrometry analysis provided 98.5% coverage, with high redundancy of overlapping peptides identified for all investigated states (Supplementary Fig. 9). Peptides spanning regions of interest were selected and mapped onto the wild-type crystal structure (PDB: 6RY4; Fig. 6). Exchanged deuterons over time were compared between the four different states (wild-type and mutant, both with and without ligand) for each peptide (Fig. 6, right side). As expected, peptides spanning the dimer interface showed reduced protection in the monomeric mutant R576E compared to wild type, both with and without ligand present (Fig. 6a). EB-47 binding resulted in prominent protection of peptides covering the NAD$^+$ binding pocket for both the wild-type and the mutant protein (Fig. 6b), including the

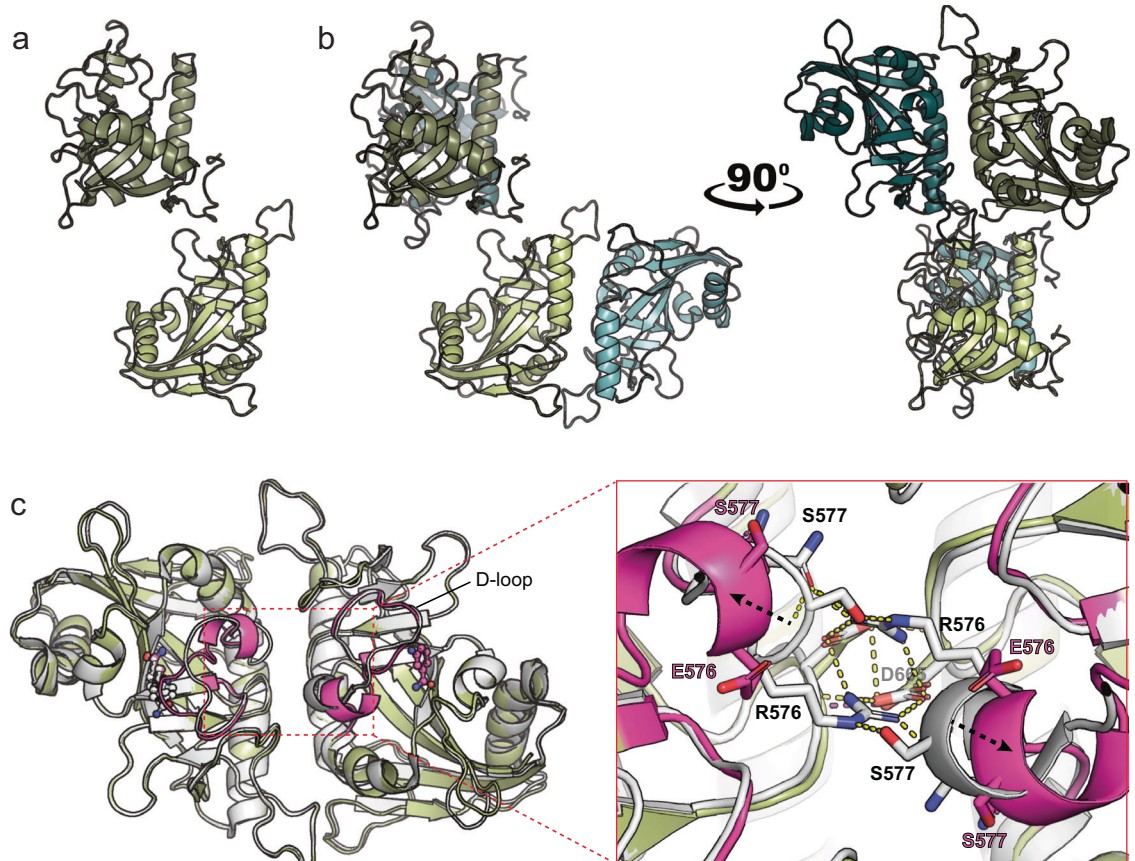

**Fig. 5 | Crystal structure unveils the effect of the R576E mutation on the interface interaction network and its role in anchoring of the D-loop. a** The asymmetric unit contains two chains (A and B) that do not engage in a dimer. **b** Each chain interfaces its symmetry mate (shades of blue), revealing chainA-chainA and chainB-chainB homodimers in the crystal lattice. **c** The R576E mutant (green) superimposed with WT PARP15 structure 6RY4 (white). The D-loop and 3AB in the R576E mutant structure are depicted in magenta. The R576E mutation prevents the N-terminus of the D-loop from being anchored through a polar bond network (yellow dashes) at the dimer interface. Black dashed arrows highlight the offset of key interface residues observed in the R576E mutant crystal structure. See also Supplementary Tables 5 and 6, and Supplementary Fig. 7.

D-loop (present in Fig. 6a peptide II and Fig. 6b peptide I). Although increased D-loop flexibility was expected based on the structural considerations discussed above, it was not the only effect of lost dimerization; we also observed important differences in segment β6-β8. The N-terminal part of this segment engages in the dimer interface (Fig. 6a,c; peptide III) and its C-terminal part represents the entry site to the active site. While being more flexible in the R576E variant, peptides in this area of the mutant protein are also unresponsive to EB-47 binding and this is in stark contrast to the wild-type protein (Fig. 6c). Loop β7-β8, the acceptor loop (Fig. 6c, peptide IV), harbors the leucine (L659) that is homologous to PARP1´s catalytic glutamate[36], protruding into the active-site pocket. We conclude from these data that dimerization of PARP15 allows, or actively induces, the conformational arrangements of the acceptor site that are necessary for catalytic activity, brought upon by a stabilization of the β6-β7 loop as it engages in the interface (Supplementary Fig. 10).

## Discussion

Despite important advances in recent years, we still know very little about structure-function relationships within the PARP family – what regulatory mechanisms govern the enzymatic activities and binding events in these medically important proteins? The most prominent exception is PARP1: Its ART domain is kept in a catalytically inactive state unless a DNA binding event is relayed throughout the multi-domain assembly to the α-helical domain (HD), which undergoes a conformational change to activate the ART domain[37–39]. A similar mechanism regulates PARP2[40,41]. Homologous helical extensions to the ART domain in PARP3[42] and PARP4[43] as well as a non-homologous α-helical subdomain in PARP16[44] and perhaps other family members[7] suggest similar mechanisms of ART domain regulation in these enzymes; but this is yet to be experimentally confirmed. Notably, all family members except PARP16 possess additional intramolecular domains, and some have known external accessory factors[4,45], providing potential for further discovery in the regulation of PARP catalytic activity and target interaction. By contrast, studies describing intrinsic regulatory mechanisms in the conserved ART domain – arguably the basis for all further regulation – have been scarce.

Here, we describe how a serendipitous observation led to the discovery that ART domain dimerization in PARP15 prepares the target acceptor site for transfer of ADP-ribose from NAD[+] in a mechanism that is indispensable for PARP15 catalytic activity. The domain forms high-affinity homodimers in solution ($K_d$ = 313 nM determined by AUC; Fig. 1k, Supplementary Table 2), forming an interface that is consistent with the one captured in all existing PARP15 crystal structures to date (Fig. 2). Interface mutants based on these structures destabilized the dimer. Loss of dimerization coincided with loss in catalytic activity (Fig. 3) without affecting the folding state or the ability to bind to small-molecule inhibitors (Supplementary Fig. 4). Mutation of the dimer interface abrogated catalytic activity in the context of both automodification and target modification in vitro (Fig. 3f and Supplementary Fig. 5b). The same interface mutations strongly compromised MARylation activity in cells and altered the localization pattern of the

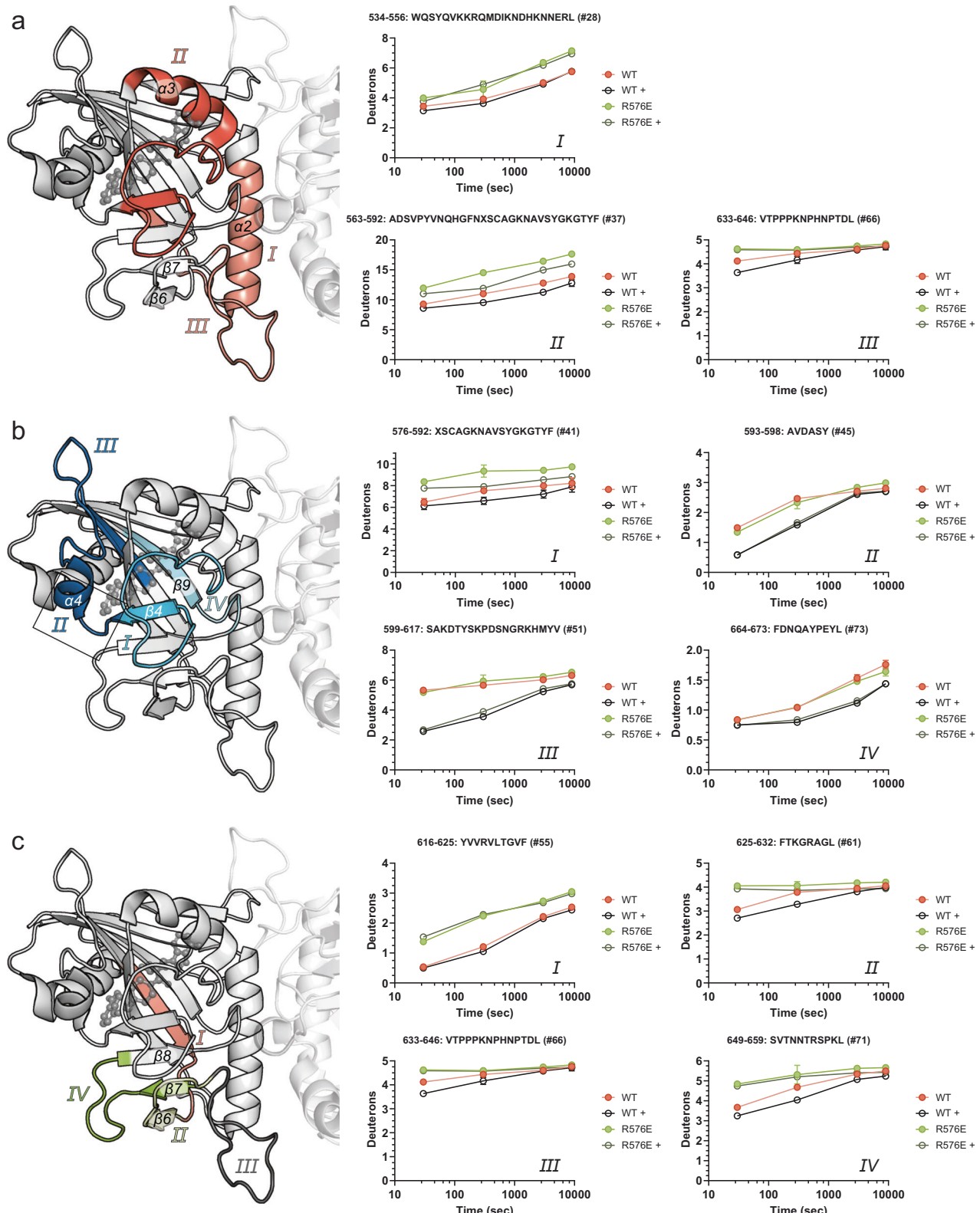

**Fig. 6 | HDX-MS provides insights into the in-solution differences between wild-type PARP15 and the monomeric R576E mutant.** The plots on the right show the number of deuterons exchanged over time for notable peptides in all four states (wild-type and R576E mutant, in the presence (+) and absence of EB-47.) Shown are means ± SD; n = number of peptide identifications per time point and state, as provided in the Source Data file. The selected peptides are colored on the PARP15 ART crystal structure (6RY4) on the left. An EB-47 molecule, based on overlay with

PARP16 structure 6HXR, is shown in ball-and-stick representation to signify the active site. **a** Peptides located at the dimer interface are more exposed in the monomeric variant compared to wild-type PARP15. **b** EB-47 binds to both PARP15 variants and protects peptides located at the active site. **c** Peptides #61, #66 and #71 located at the acceptor site respond to EB-47 binding in the wild-type protein, but not in the R576E mutant. See also Supplementary Figs. 9 and 10. Source data are provided as a Source Data file.

full-length PARP15 protein (Fig. 4 and Supplementary Fig. 6). Sub-cellular localization has already been linked to catalytic activity in PARP11 and PARP14[10,46]. Thus, our data suggests that dimerization of PARP15 is a physiological phenomenon and a pre-requisite for PARP15 activity and localization.

Homo-oligomerization is a common regulatory mechanism in proteins, but within the PARP family, it has only been observed in the form of polymerization in tankyrases[47,48]. Recent cryo-EM structures of a sterile alpha motif (SAM) domain-ART domain construct, the minimal active unit of TNKS2, revealed that the catalytic domains engage in a homotypical interaction that utilizes an interface that is homologous to the one observed in the PARP15 homodimer[49]. This interface was proposed to regulate tankyrase activation[49,50]; but contrary to PARP15, the TNKS2 dimer is restricted to the active polymer, since the ART:ART contact relies on polymerization via the SAM domain[49,50].

Systematic comparison of tankyrase crystal structures revealed that the NAD+ binding pocket opens as the TNKS ART domain engages in the interface, primarily caused by intermolecular interactions involving residues at the base of the D-loop[49,50]. Owing to the high similarity of the interfaces, we hypothesized that a similar mechanism may be present in PARP15. Due to the propensity of PARP15 to crystallize in a dimeric form, we were unable to compare truly monomeric with dimeric structures, as has been done for the tankyrases. Therefore, we crystallized the PARP15 R576E mutant to obtain a model of monomeric PARP15. However, under the conditions of the experiment, the dimer was apparently selected during crystallization, yielding a structure that was nearly identical to the wild-type protein (Fig. 5c). Nevertheless, the structure revealed that the mutation not only breaks the central salt-bridge network at the interface but also prevents multiple hydrogen bonds from forming (Fig. 5c), as corroborated by PISA analysis (Supplementary Table 6). This may also explain the milder effect on catalytic activity caused by those mutations that retain the R576 side chain (Fig. 3), because the guanidinium group makes the strongest single contribution to hydrogen bonding at the D-loop base. This polar interaction network appears to pull the D-loop towards the interface, thereby opening the NAD+ binding site in the wild-type protein, much like what is observed in TNKS2 dimeric structures, although neither the residues forming these interactions nor their positions in the structure are conserved among the two proteins.

Our ITC data did not provide any evidence for cooperativity in NAD+ binding in the PARP15 dimeric assembly (Supplementary Fig. 8). The results of our mixed-dimer experiments support this interpretation (Fig. 3f): When the NAD+ binding incompetent mutant H559Y is used to raise the concentration above the apparent dissociation constant for dimer formation, MARylation activity can proceed to saturation. Thus, we conclude that the two active sites within the PARP15 dimer function independently. However, our study provides support for a role of the D-loop in regulating active-site occupancy, where the N582 side chain has a gatekeeper role for access to the NAD+ site: Hydrogen bonding of N582 with the Y598 and Y604 side chains closes the NAD+ site. In crystal structures, this conformation is stabilized in one protomer of the wild-type dimer by contacts within the crystal lattice[26,28] preventing ligand binding in that protomer. A homologous interaction prevents ligand binding in the inactive PARP13 ART domain[26]. Mutation of Y598 in PARP15 prevents that interaction and allows ligand binding to both protomers[28]. In both protomers in our R576E structure, the N582 side chain is observed in a unique conformation: Instead of closing off the NAD+ site, N582 folds back into the D-loop and engages with the G580 backbone, allowing 3AB to occupy both active sites. We conclude that PARP15 has a flexible D-loop that can alternate between a closed and an open conformation depending on the conformation of gatekeeper N582.

The observed structural changes in the R576E mutant potentially explain a modestly reduced affinity towards NAD+, as observed in our ITC analysis; but this is insufficient cause for an abrogated catalytic activity. In-solution analysis of the conformational flexibility using HDX-MS revealed that primarily the loops between β6-β7 and β7-β8 are more flexible in the mutant and remain unresponsive towards EB-47 binding (Fig. 6). It is conceivable that dimer formation leads to a stabilization of the β6-β7 loop as it engages in the interface, which may arrange the acceptor site of PARP15 in a fashion that allows for target binding and/or catalysis. Interestingly, Pillay and colleagues also described an "ordering" of the β6-β7 loop in tankyrase structures that engaged in the "head-to-head" interface[49]. In TNKS2, as in PARP15, this loop forms intramolecular interactions with the D-loop of the same protomer. We theorize that this, perhaps together with an increased affinity for NAD+ in the dimeric state, may form the basis of the mechanism for catalytic activation in PARP15 (Fig. 7). Despite being rather distant from the tankyrases on the phylogenetic tree, PARP15 may have obtained a similar if not common mechanism to regulate its catalytic activity albeit without a need for polymer formation. While we do not observe stable PARP14 or PARP10 ART dimerization in vitro (Fig. 1b and 1c), AlphaFold3 predictions suggest that dimerization involving equivalent interfaces may be prominent in other PARP family members including PARP7, -11, and -12[51].

The remaining challenge is understanding how PARP15 dimerization may be regulated in vivo. Our data are consistent with the simplest form of mechanism; namely, that at sufficient local concentration, the domain might activate itself while maintaining the potential to be de-activated by interaction with other binding partners, post-translational modification, or re-localization to an environment with lower concentration. PARP15 - as TNKS1, PARP12, PARP13, and PARP14 - localizes to stress granules (SGs)[15], protein- and RNA-rich condensates which form in the cytoplasm when translation is halted[16]. PARP15 may dimerize upon its accumulation at SGs. As the dimeric form of PARP15 isoform-1 contains four ADP-ribose binding domains, it would contribute significantly to the multivalent interaction network that constitutes liquid-like compartments.

## Methods

### Molecular cloning

The expression vectors for the wild-type human ART domains of PARP10 (Asn819-Val1007)[19], PARP14 (His1608-Lys1801)[19] as well as WT and H599Y-PARP15 (Asn481-Ala678)[26] were generated by inserting the respective cDNAs in pNIC-Bsa4 (Genbank entry EF198106) by ligation-independent cloning. The PARP15 ART WT construct was used as template in site-directed mutagenesis (Q5 site-directed mutagenesis kit, New England Biolabs) to obtain the mutant constructs R576A, R576E, R576D, D665A, and D665R. The PARP15 macrodomain-2 construct (Gly287-Asn470) in pNIC-Bsa4 was described before[13]. A synthetic gene of PARP15 macrodomain-2 + ART ("m2-ART"; Gly287-Ala678), codon optimized for expression in E. coli, was obtained from GeneArt (Thermo Fisher Scientific) in pET151/TOPO containing an N-terminal 6xHis tag and V5 epitope. A mammalian expression vector encoding N-terminally mEGFP-tagged full-length PARP15 was used as a template for site-directed mutagenesis (Q5 site-directed mutagenesis kit, New England Biolabs) to obtain the mutant constructs H559Y, R576A, R576E, R576D, D665A, and D665R. Note that sequence numbering in our previous publications[13,26] followed an earlier gene annotation. The human SRPK2 expression plasmid was a gift from Nicola Burgess-Brown (Addgene plasmid # 39047).

### Protein expression and purification

Expression plasmids were transformed into BL21(DE3) cells (New England Biolabs). The cells were grown in terrific broth medium (Supelco) supplemented with 100 μg/ml ampicillin or 50 μg/ml kana-mycin in a LEX bioreactor (Epiphyte3) at 37 °C under constant aeration. After overnight induction at 18 °C with 0.4 mM isopropyl β-D-thiogalactopyranosid (Thermo Scientific), cells were harvested by centrifugation at 4600 rcf, 15 min. Harvested cells were resuspended in

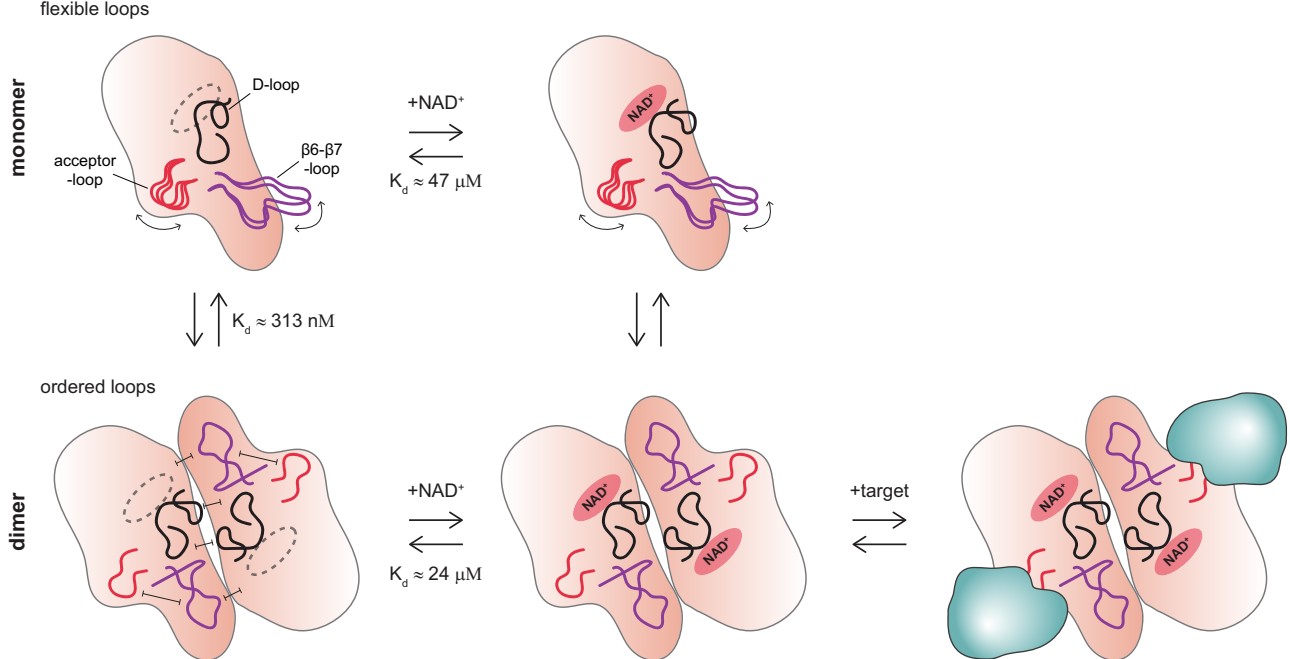

**Fig. 7 | Dimerization prepares PARP15 for target engagement by stabilizing distinct surface loops.** Working model of PARP15 activation based on the data presented in this study: ART domain dimerization and NAD$^+$ binding together bring surface loops into position for engagement of target protein and transfer of ADP-ribose. Both monomer and dimer can bind NAD$^+$ but dimerization promotes NAD$^+$ binding. $K_d$ values for ART domain dimerization and dinucleotide binding are from our experiments detailed in Fig. 1 and Supplementary Fig. 8 (values for EB-47), respectively.

lysis buffer (50 mM HEPES pH 7.5, 300 mM NaCl, 10% glycerol, 0.5 mM TCEP) supplemented with protease inhibitor cocktail (Roche) and benzonase nuclease (Millipore) and lysed by sonication. The lysates were clarified by centrifugation at 22,000 rcf for 25 min at 4 °C. The clarified lysates were filtered to 0.4 µm and loaded onto a HiTrap TALON crude column (Cytiva). Bound proteins were eluted with an imidazole gradient up to 400 mM, and peak fractions were pooled and injected into a HiLoad 16/600 Superdex 75 pg column (Cytiva) equilibrated in 50 mM HEPES pH 7.5, 300 mM NaCl, 10% glycerol, 0.5 mM TCEP. The PARP15 isoform-2 (m2-ART) construct was instead loaded onto a HiLoad 16/600 Superdex 200 pg column (Cytiva), equilibrated in 50 mM NaPO$_4$, pH 7.3, 0.5 mM TCEP, 5% glycerol. The pooled m2-ART elution fractions were subjected to cation exchange chromatography on a HiTrap SP XL column (Cytiva), and PARP15 was eluted with a shallow gradient to 1 M NaCl. For crystallography of PARP15 ART mutant R576E, the IMAC elution buffer was exchanged to 50 mM NaPO$_4$, pH 6.8, 20 mM NaCl, 10% glycerol, 1 mM TCEP. The protein was then loaded onto a HiTrap SP XL column (Cytiva) and eluted with a shallow gradient to 1 M NaCl before injection onto a HiLoad 16/600 Superdex 75 pg column (Cytiva) equilibrated in 50 mM HEPES pH 7.5, 100 mM NaCl, 10% glycerol, 0.5 mM TCEP. The purity of the proteins was evaluated by SDS-PAGE. Purified proteins were aliquoted and stored at -80 °C.

## SEC-RALS/LALS (OMNISEC)

120 µl protein solution (0.75 mg/ml PARP15 constructs, 0.35 mg/ml PARP14 ART, 0.65 mg/ml PARP10 ART) was injected in duplicate into the OMNISEC system (Malvern Panalytical) equilibrated in 50 mM HEPES pH 7.5, 300 mM NaCl, and 1 mM TCEP and operated at a flow rate of 0.5 ml/min. The OMISEC RESOLVE module was equipped with a Superdex 200 Increase 10/300 GL column (Cytiva). Right-angle and low-angle light scattering (RALS 90° angle and LALS 7° angle), differential refractive index, viscosity, and UV/VIS signal were measured by the OMNISEC REVEAL module. The integrated software (OMNISEC v11.36) was used for data collection and analysis. The refractive index

and inferred molecular weight of representative runs were plotted in GraphPad PRISM.

## Chemical crosslinking

Protein at 2, 10, or 20 µM was reacted with bis(sulfosuccinimidyl) suberate (BS3; Sigma-Aldrich) at a molar ratio of 2:1 (protein to crosslinker) in reaction buffer (50 mM HEPES pH 7.5, 100 mM NaCl, 4 mM MgCl$_2$, 0.2 mM TCEP). After 10 min incubation at room temperature, the reaction was quenched with 0.4 mM Tris pH 8 and the addition of Laemmli buffer. The proteins were resolved on 4-12% Bis-Tris gels (Invitrogen) and stained with Coomassie[52].

## Mass spectrometry to analyze chemically crosslinked samples

*In-solution digestion –* Cross-linked PARP15 ART construct was digested in solution: The pH of the samples was adjusted to 7.8 by addition of ammonium bicarbonate (ABC; Sigma-Aldrich) to a final concentration of 100 mM. The protein was reduced by the addition of 5 mM DL-dithiothreitol (DTT; Sigma-Aldrich) and incubated at 37 °C for 30 min, followed by alkylation using iodoacetamide (IAA; Sigma-Aldrich) at 12 mM and incubation in the dark for 20 min. Sequencing-grade modified trypsin (Promega, Madison, WI, USA) was added to a concentration of 1:100 (trypsin:protein) before the sample was incubated at 37 °C for four hours. A second aliquot of trypsin was added giving a final concentration of 1:50 (trypsin:protein). After overnight incubation at 37 °C formic acid (FA; Fisher Scientific) was added to a final concentration of 0.5 %.

*In-gel digestion –* Gel bands were excised from SDS-PAGE gels, cut into 1 × 1 mm pieces and transferred to 1.5 ml tubes. The gel pieces were washed and incubated for 30 min three times with 500 µl 50% acetonitrile (ACN, VWR), 50 mM ABC. The gel was dehydrated using 100% ACN before the proteins were reduced with 25 µl 10 mM DTT in 50 mM ABC for 30 min at 37 °C. DTT was removed, and the gel was dehydrated again using 100% ACN before the proteins were alkylated with 25 µl 55 mM iodoacetamide in 50 mM ABC for 30 min in the dark at room temperature. The gel was dehydrated one last time with 100 %

ACN before the proteins were digested by adding 25 µl 12 ng/µl trypsin (sequence grade modified porcine trypsin, Promega, Fitchburg, WI, USA) in 50 mM ABC and incubated on ice for 4 hours before 20 µl 50 mM ABC was added and the proteins were incubated overnight at 37 °C. The following day, formic acid was added to a final concentration of 0.5 %, to get a pH of 2-3, before the peptide solutions were extracted and transferred into new tubes.

*LC-MSMS* – The peptides were cleaned up using C18 reversed-phase micro columns with a 2% ACN, 0.1% FA equilibration buffer and an 80% ACN, 0.1% FA elution buffer. The collected samples were dried in a fume hood and resuspended in 15 µl 2% ACN, 0.1% FA. The samples were then injected into an ultra-high-pressure nanoflow chromatography system (nanoElute, Bruker Daltonics). The peptides were loaded onto an Acclaim PepMap C18 (5 mm, 300 µm id, 5 µm particle diameter, 100 Å pore size) trap column (Thermo Fisher Scientific) and separated on a Bruker Pepsep Ten C18 (75 µm × 10 cm, 1.9 µm particle size) analytical column (Bruker Daltonics). Mobile phase A (2% ACN, 0.1% FA) was used with the mobile phase B (0.1% FA in ACN) for 45 min to create a gradient (from 2 to 17% B over 20 min, from 17 to 34% B over 10 min, from 34 to 95% B over 3 min, at 95% B over 12 min) at a flow rate of 400 nl/min and a column oven temperature of 50 °C. The peptides were analysed on a quadrupole time-of-flight mass spectrometer (timsTOF Pro, Bruker Daltonics), via a nano electrospray ion source (Captive Spray Source, Bruker Daltonics) in positive mode, controlled by the OtofControl 5.1 software (Bruker Daltonics). The temperature of the ion transfer capillary was 180°. A DDA method was used to select precursor ions for fragmentation with one TIMS-MS scan and 10 PASEF MS/MS scans. The TIMS-MS scan was acquired between 0.60–1.6 V s/cm² and 100–1700 m/z with a ramp time of 100 ms. The 10 PASEF scans contained a maximum of 10 MS/MS scans per PASEF scan with a collision energy of 10 eV. Precursors with a maximum of 5 charges with an intensity threshold to 5000 a.u. and a dynamic exclusion of 0.4 s were used. The mass spectrometry data have been deposited to the ProteomeXchange Consortium via the PRIDE partner repository[53] with the dataset identifier PXD067296.

*Data analysis* – Raw data were processed using Mascot Distiller (version 2.8) and searched using Mascot Deamon (version 2.8) against an in-house database containing the PARP15 protein sequence using the following settings: Precursor ion tolerance 8 ppm, MS/MS fragment mass tolerance 0.015 Da, trypsin as protease, 1 missed cleavages site, methionine oxidation as variable modification, and carbamidomethylation as fixed modification. The obtained peak lists were analysed in xiSEARCH[23]. As both in-solution and in-gel digest data obtained similar results, three peak lists of each method were pooled and analysed together to improve stringency of the results. The following parameters were applied: crosslinker = BS3(small scale); enzyme = trypsin; miscleavages = 3; fixed modifications = carbamidomethylation (C); variable modifications = oxidation (M); variable (linear peptides) = BS3 amidated, BS3 hydrolyzed; ions = b-ion, y-ion; peptide tolerance 6 ppm; fragment tolerance = 20 ppm. The xiSEARCH default was applied for BS3 linkage specificity (lysine, serine, threonine, and tyrosine). The output was visualized in xiVIEW[54], omitting low-scoring peptides.

## Mass photometry

The concentration-dependent dimerization of PARP15 constructs was followed on a TwoMP mass photometer (REFEYN, Oxford, UK). Microscopy glass slides (24 × 50 mm) were alternately cleaned with ultrapure water and isopropanol and finally dried under a clean nitrogen stream. Silicone gasket wells (Grace Biolabs, Merck Life Science AB, Solna, Sweden) were placed on the clean glass slide, and the focus was adjusted to a well containing 15 µl sample buffer (20 mM HEPES pH 7.5, 50 mM NaCl, 4 mM MgCl₂, 0.5 mM TCEP). The proteins were added to the sample buffer at varying concentrations (see Fig. 1h, i) and landing events were recorded over 60 seconds in a frame size of

900 × 354 pixels and at a frame rate of 496 Hz. The DiscoverMP software provided by the manufacturer was used for data analysis. To translate recorded mass photometry contrast into molecular weight, a native PAGE marker (NativeMark™ Unstained Protein Standard; Invitrogen) was used for calibration.

## Analytical ultracentrifugation

To determine whether auto-MARylation of PARP15 might affect its oligomeric state, 40 µM PARP15 ART was incubated with 100 µM NAD⁺ (Roche) in 50 mM HEPES pH 7.5, 100 mM NaCl, 0.2 mM TCEP, 4 mM MgCl₂ for 1 h at ambient temperature. Unreacted NAD⁺ was removed by buffer exchange on a HiTrap Desalting column (Cytiva) equilibrated in AUC buffer (20 mM Sodium Phosphate, pH 7, 150 mM NaCl), and an unmodified ART sample was exchanged into AUC buffer in the same way. Both samples were finally diluted to 9 µM in AUC buffer. Sedimentation velocity experiments were carried out in an Optima analytical ultracentrifuge equipped with an AN-60Ti rotor (Beckman Coulter) and measured in intensity mode at 220 nm and 280 nm. Samples were centrifuged for either 12 h at 40,000 rpm, or 5 h at 58,000 rpm. Epon 2-channel centerpieces were used for speeds of 40,000 rpm, SedVel60k centerpieces for 58,000 rpm.

Samples from the auto-MARylation experiment were recovered from the AUC cells after the run and were subjected to Western blotting using an anti-MAR specific antibody (see below).

UltraScan III (ver. 4) was used for data analysis. Triplets with insufficient signal were filtered out. All remaining data sets were first fitted by two-dimensional spectrum analysis (2DSA), fitting time- and radially invariant noise as well as meniscus and bottom position. Enhanced van Holde-Weischet analysis was performed, and the individual integral distributions were overlaid in GraphPad PRISM to visualize monomer-dimer equilibria. The dataset collected at 58,000 rpm (0.5 µM PARP15, no NAD⁺ treatment; measurement at 220 nm) was used for discrete model genetic algorithm (DMGA) analysis to determine the dissociation constant of homodimerization. During fitting, molecular weights determined based on the protein sequence were fixed, and partial specific volume and frictional ratio, as well as the dissociation constant and rate constant, were floated. The extinction coefficient at 220 nm was determined from a global fit of spectral scans measured over a range of PARP15 dilutions using the Spectral Fitter function in UltraScan III. 100 Monte Carlo iterations were performed to determine 95% confidence intervals for each parameter.

## Bioinformatic analysis and structure visualization

AlphaFold3[32] was used for predictions of the PARP15 m2-ART structure and PARP15 ART homodimer. Crystal structures of the PARP15 ART domain deposited in the Protein Data Bank (PDB) were analyzed using the PISA server[30]. PyMOL[55] was used for structural alignments and general representation of experimental structures and computational models.

## Differential Scanning Fluorimetry (nanoDSF)

The PARP15 ART domains were diluted to 0.2 mg/ml in 50 mM HEPES pH 7.5, 100 mM NaCl, 4 mM MgCl₂, 0.2 mM TCEP. 1 mM 3-aminobenzamide (Sigma-Aldrich) was added, and the samples were incubated at room temperature for at least 45 min. Three Prometheus NT.48 standard capillaries (NanoTemper) were loaded per sample, and the intrinsic fluorescence at 330 nm and 350 nm was recorded over a temperature gradient of 25–95 °C (+1 °C/min) in a Prometheus Pantha (NanoTemper). Raw data, first derivatives, and melting temperatures were exported from the PR.ThermControl software and plotted in GraphPad PRISM.

## Analytical size exclusion chromatography

A Superdex 200 Increase 10/300 GL column was equilibrated in running buffer (50 mM HEPES pH 7.5, 300 mM NaCl, 10% glycerol, 0.5 mM TCEP)

and calibrated using ovalbumin (44 kDa, 2.5 mg/ml), conalbumin (75 kDa, 2.5 mg/ml), aldolase (158 kDa, 2.5 mg/ml), ferritin (440 kDa, 2.5 mg/ml), thyroglobulin (669 kDa, 2.5 mg/ml), RNase (13.7 kDa, 3.2 mg/ml), Blue Dextran (2.5 mg/ml) (Gel Filtration Calibration kit; Cytiva). The column was operated at a flow rate of 0.7 ml/min on an ÄKTA Pure FPLC (Cytiva). Subsequently, 100 μl of each PARP15 ART construct at 0.7 mg/ml was injected into the column, using the same run parameters as before. Chromatograms were analyzed in Microsoft Excel and re-plotted using GraphPad PRISM.

### Analysis of MARylation levels by Western blot (in vitro)

Pre-ADP-ribosylation of PARP15 ART domains (wild-type, H559Y mutant, R576E mutant) during expression in *E. coli* was assessed by western blotting. The proteins were diluted to 10 μM into assay buffer (20 mM HEPES pH 7.5, 50 mM NaCl, 4 mM MgCl$_2$, 0.2 mM TCEP). Each sample was split in two, and 50 μM NAD$^+$ (Roche) was added to one set of the reaction. NAD$^+$ was omitted in the other set (control). All reactions were incubated at 37 °C, and Leammli buffer was added after 10, 20, or 30 min (only timepoint for the control), followed by heating for 1 min at 95 °C before subjection to gel electrophoresis and western blotting (see below).

For auto-modification assays, 10 μM of PARP15 ART constructs, or 5 μM in case of mutant combinations, were used. For trans-modification assays, 1 μM of PARP15 ART constructs and 5 μM of SRPK2 were used. The proteins were diluted in assay buffer (20 mM HEPES pH 7.5, 50 mM NaCl, 4 mM MgCl$_2$, 0.2 mM TCEP) and incubated for 30 min at 37 °C, 250 rpm, prior to the addition of NAD$^+$. 50 μM NAD$^+$ containing 10% biotin-NAD$^+$ (Tocris Bioscience) was added to start the ADP-ribosylation reaction, followed by incubation for another 30 min. The reactions were stopped by the addition of Laemmli buffer and heating for 1 min at 95 °C.

Samples were then loaded onto NuPage 4-12% Bis-Tris gels (Invitrogen) and resolved by electrophoresis. Proteins were transferred from the gel onto a PVDF membrane (Millipore) in a wet-transfer set-up, using a transfer buffer consisting of 3 g/L Trisma-Base, 14.4 g/L glycine, and 10% methanol. The transfer was performed for 60 min at 25 V. Ponceau S (Sigma-Aldrich) staining of the membrane was used to assess transfer efficiency and served as a loading control. For reactions performed with spiked-in biotin-NAD$^+$, the membrane was blocked with 1% BSA (Sigma-Aldrich) in TBS-T (Tris-buffered saline, 0.1% Tween-20) for 20 min and then incubated in 0.5 μg/ml Streptavidin-HRP (Pierce) in 1% BSA in TBS-T buffer for 35 min. For reactions without biotin-NAD, the membrane was blocked with 5% milk in TBS-T buffer and then incubated in primary antibody (Mono-ADP-Ribose, Bio-Rad, HCA354; 2 μg/ml) at 4 °C overnight, followed by secondary antibody (goat anti-rabbit - Peroxidase, Sigma, A8275; 1:10,000) for 2 h at ambient temperature. For detection, we used Clarity Western ECL substrate (Bio-Rad) following the manufacturer's instructions. Chemiluminescent and colorimetric images were recorded on a ChemiDoc Imaging System (Bio-Rad).

### Analysis of MARylation levels by fluorescent protein overlay (MacroGreen)

In vitro ADP-ribosylation activity of PARPs was determined using a green fluorescent protein fusion of a modified Af1521 macrodomain[33]. For auto-ADP-ribosylation assays, a 2:1 dilution series (two parts protein, one part buffer) in assay buffer (50 mM HEPES pH 7.5, 100 mM NaCl, 4 mM MgCl$_2$, 0.2 mM TCEP) starting from 1 μM PARP protein was prepared in a 96-well flat-bottom plate. The plate was incubated for 20 min at room temperature to allow for a monomer-dimer equilibrium to be reached. Reactions were started by addition of 120 μM NAD$^+$. ADP-ribosylation reactions were performed at room temperature for 30 min under constant shaking at 250 rpm. The content of each well was distributed into three individual wells on a MaxiSorp™ plate (Thermo Scientific). The proteins were allowed to bind to the

plate for an additional 15 min, followed by three washes with TBS-T (Tris-buffered saline, 0.1% Tween-20) to remove unreacted NAD$^+$. Wells were blocked for 15 min with 1% BSA in TBS-T (at room temperature, 250 rpm), then rinsed once with TBS-T.

For comparison of the activity of PARP15 interface mutants to WT PARP15, mother reactions containing 1 μM PARP15 variants, or combinations of 0.5 μM PARP15 variants, were prepared in assay buffer and incubated for 20 min at room temperature, 250 rpm. The reactions were started by the addition of 100 μM NAD$^+$, and incubated at ambient temperature for 30 min, 250 rpm. The reactions were distributed into four wells on a MaxiSorp™ plate (Thermo Scientific), creating quadruplets. Further procedures were as above.

When PARP15 macrodomain-2 was used as a target, 250 nM PARP15 ART WT, D665A, or R576A, was mixed with titrations of PARP15 ART H559Y in 50 mM HEPES pH 7.5, 100 mM NaCl, 4 mM MgCl$_2$, 0.2 mM TCEP (2-fold dilution series of H559Y starting from 2 μM). PARP15 ART H559Y was omitted in the control reaction and another set of reactions containing only H559Y was set up to show that the mutant is indeed inactive. An equilibrium was allowed to establish for 1 h at room temperature and 250 rpm shaking. Meanwhile, the wells of a MaxiSorp™ plate (Thermo Scientific) were coated with PARP15 macrodomain-2 for 30 min at room temperature, 250 rpm, and subsequently blocked with 1% BSA in reaction buffer for an additional 15 min. The wells were rinsed twice with 150 μl reaction buffer before 50 μl of the PARP15 ART-containing samples were added to the MaxiSorp™ plate (Thermo Scientific) and the ADP-ribosylation reactions were started by the addition of 100 μM NAD$^+$. The reaction was allowed to proceed for 40 min at room temperature, 250 rpm. Further procedures were as above.

For detection, 1 μM MacroGreen was added to the wells. Fluorescence intensity was measured in a CLARIOStar multimode reader (BMG Labtech) using a 470-15 nm extinction filter and a 515-20 nm emission filter. Raw data were plotted in GraphPad PRISM.

### Mammalian cell culture and immunofluorescence

HEK 293T (ATCC, CRL-3216) cells were grown in DMEM medium (Gibco), supplemented with 10% fetal bovine serum (FBS; Gibco) and 1x GlutaMAX™ Supplement (Gibco) at 37 °C and 5% CO$_2$. On the day before transfection, the cells were seeded onto Poly-D-Lysine-coated coverslips. 0.2 μg of plasmid DNA (N-terminally mEGFP tagged full-length PARP15 variants; wild-type, H559Y, R576A, R576D, R576E, D665A, and D665R) was used for transfection with jetOPTIMUS® DNA transfection Reagent (Polyplus-transfection). After 4 h of incubation, the media were replaced by fresh media supplemented with 100 nM PARP1 inhibitor Saruparib (AZD5305; final concentration of 0.1% DMSO; Selleck), followed by overnight incubation. PARG inhibitor (PDD 00017273; Sigma-Aldrich) was added at a final concentration of 1 μM. After 4 h, the cells were washed with cold 1x phosphate-buffered saline (PBS) and fixed in 3.7% PFA in 1x PBS for 10 min at ambient temperature. The cells were permeabilized in 0.5% Triton-X100 in 1x PBS and blocked for 1 h in 2% BSA, 0.1% Triton-X100 in 1x PBS. Anti-ADP-ribose specific antibody (Cell Signaling Technology) was added at a 1:1000 dilution in 2% BSA, 0.1% Triton-X100 in 1x PBS. After 2 h at ambient temperature, the cells were washed three times in 1x PBS. Secondary antibody (Donkey Anti-rabbit-Alexa568) was added at 1:1000 dilution in 2% BSA, 0.1% Triton-X100 in 1x PBS and incubated for 1 h at ambient temperature. Hoechst stain (Invitrogen) was added for 5 min at the end of the incubation with secondary antibody. The coverslips were mounted onto glass microscopy slides with ProLong™ Gold Antifade Mountant (Invitrogen). Immunofluorescence was imaged on a Zeiss Apotome.2 on an Axio Imager with a 20x air objective. Image analysis was carried out in Fiji (ImageJ) software. Green (GFP) and Alexa568 (ADPr) channel brightness and contrast settings were applied over all micrographs for direct comparison between mutants. Nuclear masks derived from Hoechst staining were

applied to green and red channels, to obtain nuclear GFP and ADPr signals for scatter plots. For violin plots, GFP-negative nuclei were removed based on intensity thresholding. A Kruskal-Wallis Test paired with Dunn's multiple comparisons test was carried out in Graphpad PRISM to determine significance values. Asterisks indicate *P*-values; **$p \le 0.01$, ****$p \le 0.0001$.

## Cellular ADP-ribosylation assay (western blot)

HEK 293 T cells (ATCC, CRL-3216) were transfected and treated with PARP1 inhibitor Saruparib (AZD5305; Selleck) and PARG inhibitor (PDD 00017273; Sigma-Aldrich) exactly as described in the materials and methods section "Mammalian cell culture and immunofluorescence". The media was aspirated, the cells were washed in cold 1x PBS and frozen at −80 °C. Cells were lysed in ice-cold lysis buffer (50 mM HEPES pH 7.8, 150 mM NaCl, 1 mM $MgCl_2$, 1% Triton-X100, 1 mM TCEP) supplemented with protease (Pierce) and phosphatase inhibitor cocktail (Sigma-Aldrich) and 30 μM pan-PARP inhibitor Phthal 01[56]. Cell lysates were clarified by centrifugation at 14,000 rcf for 10 min at 4 °C. Total protein concentration for each sample was determined via Bradford assay (Bio-Rad) and subsequently adjusted with lysis buffer to obtain 2 mg/ml. Samples were mixed with Laemmli buffer (10% glycerol, 50 mM Tris-Cl (pH 6.8), 2% SDS, 1% β-mercaptoethanol, 0.02% bromophenol blue) and boiled at 95 °C for 5 min before loading onto 10% SDS-PAGE gels. Electrophoresis was performed at 165 V for 70 min. Transfer onto nitrocellulose membranes was performed in a Trans-Blot Turbo Transfer System (Bio-Rad) in High Molecular Weight mode (10 min). Successful transfer was tested by Ponceau S staining. The membranes were blocked in 5% milk (Carnation) in PBS-T (Phosphate-buffered saline, 0.1% Tween-20) for 1 h, then washed in PBS-T three times. Primary antibodies were diluted in PBS-T, 2% BSA, 0.05% $NaN_3$ (anti-GFP (Chromotek): 1:1000, anti-tubulin (Cell Signaling Technology): 1:2000, anti-mono/poly-ADP-ribose (Cell Signaling Technology): 1:1000, anti-mono-ADP-ribose (Bio-Rad): 2 μg/ml) and added to the membranes for overnight incubation at 4 °C. The membranes were washed three times in PBS-T before the addition of HRP-conjugated secondary antibody (Goat anti-rabbit (Invitrogen): 1:10,000; goat anti-mouse (Invitrogen): 1:5,000; in 5% milk in PBS-T), followed by incubation for 1 h at ambient temperature. The western blots were developed with SuperSignal™ West Pico PLUS Chemiluminescent Substrate (Thermo Scientific) and imaged in a ChemiDoc Imaging System (Bio-Rad).

## X-ray crystallography

Crystallization conditions for PARP15[481-678] R576E were found using a Nucleix crystal screen (Qiagen) in a vapor diffusion sitting drop set-up at 4 °C. After about 10 days, the protein crystallized in a drop containing 0.15 μl protein (10.5 mg/ml including 4 mM 3-aminobenzamide) and 0.15 μl reservoir solution (50 mM HEPES pH 7.5, 20 mM $MgCl_2$, 1 mM spermine-HCl, 5 % w/v PEG 8000). A protein crystal was placed in a cryo-solution (40 mM HEPES pH 7.5, 16 mM $MgCl_2$, 0.8 mM spermine-HCl, 4% w/v PEG 8000, 17.4% glycerol) for 2 s, then frozen in liquid nitrogen.

Diffraction data were collected at 0.97625 Å wavelength at the BioMAX beamline at MAX IV Laboratory in Lund, Sweden, which was equipped with a DECTRIS Eiger 16 M detector. 3600 images were collected with a rotation range of 0.1° per image. Reflections were integrated, indexed, and scaled with the automatic data processing pipeline autoPROC (XDS, AIMLESS)[57–59], incorporating anisotropic data reduction using STARANISO[60]. The structure was solved by molecular replacement in Phaser-MR, using chain A of the PARP15 wild-type crystal structure (PDB: 7OTF) as a search model. The structure was refined with riding hydrogen atoms in phenix.refine[61], and model building was performed in Coot[62]. Details on crystallographic data collection and structure refinement statistics are given in Supplementary Table 5.

## Isothermal Titration Calorimetry

EB-47 (Sigma-Aldrich) was dissolved at a concentration of 10 mM in water. PARP15 WT and R576E were transferred to ITC buffer (20 mM HEPES pH 7.5, 100 mM NaCl, 4 mM $MgCl_2$, and 0.2 mM TCEP) on PD-10 columns and diluted to 44 μM in the same buffer. Experiments were performed using an ITC200 instrument (MicroCal). Each measurement included 13 injections (3 μl) of ligand (duration 6 seconds) to the protein in the cell (200 μl) at 25 °C with an equilibration time of 150 s and a stirring speed of 1000 rpm. Results were analyzed with the MicroCal PEAQ-ITC Analysis Software using the One Set of Binding Sites fitting model.

## Hydrogen-Deuterium Exchange Mass Spectrometry (HDX-MS)

*Experiment* – pH measurements were made using a SevenCompact pH-meter equipped with an InLab Micro electrode (Mettler-Toledo), a 4-point calibration (pH 2,4,7,10) was made prior to all measurements. The HDX-MS analysis was made using automated sample preparation on a LEAP H/D-X PAL™ platform interfaced to an LC-MS system, comprising an Ultimate 3000 micro-LC coupled to an Orbitrap Q Exactive Plus MS. HDX was performed on 2 mg/ml PARP15 isoform 2 (m2-ART) in 50 mM TBS, pH 7.4, or 10 μM PARP15 ART (WT or R576E; in absence and presence of 5 mM EB-47) in 40 mM HEPES, 100 mM NaCl, 4 mM $MgCl_2$, pH 7.5. For isoform-2, for each timepoint, 3 μl sample was diluted with 27 μl labeling buffer (50 mM TBS) prepared in $D_2O$ (Sigma-Aldrich), $pH_{(read)}$ 7.0. For ART constructs, 5 μl of sample was diluted with 45 μl labeling buffer (40 mM HEPES, 100 mM NaCl, 4 mM $MgCl_2$) prepared in $D_2O$, $pH_{(read)}$ 7.1. The HDX labeling was carried out for $t = 0$, 30, 300, 3000 and 9000 s at 20 °C (isoform 2) or 18 °C (ART domain). The labeling reaction was quenched by dilution of 30 μl labeled sample with 30 μl of 1% TFA, 0.4 M TCEP, 4 M urea, pH 2.5 at 1 °C, 60 μl (isoform 2) or 55 μl (ART domain) of the quenched sample was directly injected and subjected to online pepsin digestion at 4 °C (in-house immobilized pepsin column, 2.1 ×30 mm). Online digestion and trapping were performed for 4 minutes using a flow of 50 μl/min 0.1 % formic acid, pH 2.5. The peptides generated by pepsin digestion were subjected to online SPE on a PepMap300 C18 trap column (1 mm x 15 mm) and washed with 0.1% FA for 60 s. Thereafter, the trap column was switched in-line with a reversed-phase analytical column, Hypersil GOLD, particle size 1.9 μm, 1 × 50 mm, and separation was performed at 1 °C using a gradient of 5-50 % B over 8 minutes and then from 50 to 90% B for 5 minutes, the mobile phases were 0.1% formic acid (A) and 95% acetonitrile/0.1 % formic acid (B). Following the separation, the trap and column were equilibrated at 5% organic content until the next injection. The needle port and sample loop were cleaned three times after each injection with mobile phase 5% MeOH/0.1% FA, followed by 90% MeOH/0.1% FA and a final wash of 5% MeOH/0.1% FA. After each sample and blank injection, the pepsin column was washed by injecting 90 μL of pepsin wash solution 1% FA/4 M urea/5% MeOH. To minimize carry-over, a full blank was run between sample injections. Separated peptides were analysed on a Q Exactive Plus MS, equipped with a HESI source operated at a capillary temperature of 250 °C with sheath gas 12, Aux gas 2, and sweep gas 1 (au). For HDX analysis, MS full scan spectra were acquired at 70 K resolution, AGC 3e6, Max IT 200 ms and scan range 300−2000. For the identification of generated peptides, separate undeuterated samples were analysed using data-dependent MS/MS with HCD fragmentation. A summary of the HDX experimental detail is reported in the Source Data file. The mass spectrometry data have been deposited to the ProteomeXchange Consortium via the PRIDE partner repository[53] with the dataset identifier PXD061403.

*Data analysis* – PEAKS Studio X Bioinformatics Solutions Inc. (BSI, Waterloo, Canada) was used for peptide identification after pepsin digestion of undeuterated samples. The search was done on a FASTA file with the relevant PARP15 sequence, search criteria were a mass error tolerance of 15 ppm and a fragment mass error tolerance of 0.05 Da, allowing for fully unspecific cleavage by pepsin. Peptides

identified by PEAKS with a peptide score value of log $P > 25$ and no modifications were used to generate a peptide list containing peptide sequence, charge state, and retention time for the HDX analysis. HDX data analysis and visualization were performed using HDExaminer, version 3.4.2 (Sierra Analytics Inc., Modesto, US). The analysis was made on the best charge state for each peptide, allowed only for EX2, and the first two residues of a peptide were assumed to be unable to hold deuteration. Due to the comparative nature of the measurements, the deuterium incorporation levels for the peptic peptides were derived from the observed relative mass difference between the deuterated and non-deuterated peptides without back-exchange correction using a fully deuterated sample[63]. As a full deuteration experiment was not done, full deuteration was set to 75% of the maximum theoretical uptake. The deuteration data presented are the average of all high and medium confidence results. The allowed retention time window was ± 0.5 minute. The spectra for all time points were manually inspected; low-scoring peptides, obvious outliers, and any peptides where retention time correction could not be done consistently were removed. As bottom-up labeling HDX-MS is limited in structural resolution by the degree of overlap of the peptides generated by pepsin digestion, the peptide map overlap is shown for each respective state in Supplementary Figs. 2 and 9.

### Statistical analysis

Datasets for statistical analysis were tested for normality using GraphPad Prism's Normality and Lognormality Tests, which include the D'Agostino & Pearson test, the Anderson-Darling test, the Shapiro-Wilk test, and the Kolmogorov-Smirnov test. Homogeneity of variances was assessed using the Brown-Forsythe test. For normally distributed data, an ordinary one-way ANOVA with Dunnett's multiple comparisons test was performed. For non-normally distributed data, a Kruskal-Wallis test with Dunn's multiple comparisons test was used.

### Reporting summary

Further information on research design is available in the Nature Portfolio Reporting Summary linked to this article.

## Data availability

HDX-MS and crosslinking-MS data have been deposited in the PRIDE database under accession codes PXD061403 and PXD067296, respectively. The crystallographic structure factors and atomic coordinates for the PARP15 catalytic domain mutant (R576E) in complex with 3-aminobenzamide generated in this study have been deposited in the Protein Data Bank under accession code 9IFV. Previously published structural data used in this study are available under Protein Data Bank accession codes 2BQ0, 6HXR, 6RY4, and 7OTF. Source Data are provided with this paper. All additional data are available within the main text or the Supplementary Figs. and tables. Source data are provided with this paper.

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

## Acknowledgements

This article is dedicated to the memory of Ann-Gerd ("Aja") Thorsell. We thank Céleste Sele at the Lund University Protein Production Platform (LP3) Sweden for OMNISEC analyses and assistance with nanoDSF, Emilia Strandback and Tomas Nyman (Karolinska Institutet Protein Science Core Facility, Sweden) for ITC, Anders Hofer and Saber Anoosheh (Umeå University, Sweden) for their introduction to mass photometry, Borries Demeler (University of Lethbridge, Canada) for advice on AUC data analysis, and Daniel S. Bejan and Jonathan Tullis (Oregon Health and Science University, USA) for their help with cell culture and immunofluorescence microscopy. We acknowledge MAX IV Laboratory (Lund, Sweden) for providing synchrotron radiation beamtime and support for X-ray diffraction data collection under proposal 20220149. Research conducted at MAX IV, a Swedish national user facility, is supported by the Swedish Research Council, the Swedish Governmental Agency for Innovation Systems, and Formas. This work was supported by grants 2019–04871 from the Swedish Research Council, 20-0918Pj and 23-3027Pj from the Swedish Cancer Foundation, 2021/1058 from the Crafoord Foundation, LU2022-0071 from the IngaBritt and Arne Lundberg's Research Foundation and support from the infrastructure funds of Lund University Faculty of Science (to H.S.); and grant 2R01NS088629 from the National Institutes of Neurological Disorders and Stroke (to M.S.C.). Support from the Swedish National Infrastructure for Biological Mass Spectrometry (BioMS) and the SciLifeLab, Integrated Structural Biology platform is gratefully acknowledged.

## Author contributions

C.E.: Conceptualization, experiment, data analysis, visualization, manuscript writing. A.G.G.S.: Experiment, data analysis. S.E.: Experiment, data analysis. K.B.: Experiment, data analysis. M.M.: Experiment, data collection. D.T.L.: Data analysis. M.S.C.: Conceptualization, provided key resources, and funding acquisition. H.S.: Conceptualization, manuscript writing, funding acquisition.

## Funding

## Competing interests

The authors declare no competing interests.
