## [Transparent Peer Review file · Nature Communications]

Regulation of ADP-ribosyltransferase activity by ART domain dimerization in PARP15.

Corresponding Author: Dr Herwig Schüler

Version 0:

Reviewer comments:

Reviewer #1

(Remarks to the Author)

The paper describes the significance of the dimer formation PARP15, which is a mono-ADP-ribosyltransferase that targets an unknown set of proteins as well as RNA. PARP-15 dimerizes in solution. The author concluded that there is a regulatory mechanism by which dimerization is linked to correct target engagement rather than NAD co-substrate binding and by which the two protomers of the dimer operate independently of one another. PARP15 regulation is significant and interesting as PARP's topic. The dimerization regularization is possible and interesting. However, it is important to identify and reveal the complex structure of PARP15 and its substrate to reveal the regularization. In this meaning, the paper is not enough in Nature Communication.

Major points

The activity was checked by auto-MARylation. Does the author think that there is a similar mechanism between the auto-MARylation and real substrate MARylations?

Is it necessary to use real substrate to understand the PARP15 ADP-ribosylation?

Finally, I request the figure how the dimerization works for the auto-MARylation and real substrate MARylations. The figure answers my question, including the auto-MARylation description below.

Auto-MARylation means that the PARP ADP-ribosylates itself.

If dimer exists (AB), does one protomer (A) ADP-ribosylate the same protomer (A)? If dimer exists, one protomer (A) ADP-ribosylates another protomer (B)? Alternatively, AB dimer ADP-ribosylates other dimer (CD)?

I would like to know what does the author think about auto-MARylation.

The author concluded that there is a regulatory mechanism by which dimerization is linked to correct target engagement rather than NAD co-substrate binding and by which the two protomers of the dimer operate independently of one another.

>

How does the author conclude as "dimerization is linked to correct target engagement"?

Minor points

we treated proteins with BS3, a chemical crosslinker with a spacer arm length of 11.4 Å

Please add what reaction occurs using BS3.

Fig2E

Enlarge E (Amino acids). They are too small. What is the inter-protomer peptide in yellow?

The purple line shows the inner-protomer peptide. Add the explanation in the legend.

Moreover, in this figure, it is better to explain the mutational residue used in this paper.

R576, D665, and H559. Add the explanation of the role of these residues based on the PARP structure as a supplemental figure.

Fig.3CD

C: This shows auto-MARylation of PARP15.

D: What is a fluorescent macrodomain overlay assay?

Fig3F

I understand the assay macrodomain-2 as substrate. However, I do not understand the meaning why the author used H559Y. Please insert the graphic figure as a supplemental figure.

(p19 L9~L20:

When PARP15 macrodomain2 was used as a substrate~0.2mM TCEP.

I do not understand “ was mixed with titrations of PARP15 ART H559Y”.)

Fig.4

Four hours post-transfection, cells were treated with Saruparib (AZD5305) to diminish PARP1 contribution to ADP-ribosylation levels, and 4 h before fixation, cells were treated with the PARG inhibitor PDD00017273, which we empirically determined enhances PARP15 MARYlation in cells.

>

Saruparib inhibits PARP1. Does Saruparib not inhibit other PARPs, including PARP5?

What is the mechanism to enhance PARP15 using PARG inhibitor?

Fig.5CD

Show the D-loop in the figure. (Base of the D-loop?)

P10 I24

We hypothesized ~>

Add the explanation of NAD binding cooperativity.

If NAD binds one of the protomers, the subsequent NAD binding is easy to bind to the other monomer in the dimer. The author analyzed cooperativity from the bottom figure. N is the cooperativity from the hill plot? Please add the explanation.

Why does the ART domain dimer work better for the substrate? Please explain a possible mechanism. Again, it is better to depict in the final figure (How do the dimer(mac1-mac2-ART) and substrate form the complex and express the activity). These figures could explain the D-loop interaction of the dimer and the opening for the NAD binding site, as the author suggested (p14 I8 and p14 I11~I27).

P15 I8~I9

Four ADP-ribose binding domains> The dimer includes two ART domains.

I expect future studies of the complex structure PARP15 and the substrate.

Reviewer #2

(Remarks to the Author)

The manuscript "Regulation of ADP-ribosyltransferase activity by ART domain dimerization in PARP15" submitted by Ebenwaldner et al. provides valuable insights into PARP15's protein structure, thereby aiming to elucidate catalytic mechanisms in vitro and in vivo.

The authors apply various structural biology technologies such as crystallography and HDX-MS to verify the dimerization of PARP15. Additionally, they characterize the influence of dimerization on the catalytic properties of PARP15, mostly in vitro. The data provide evidence that dimerization is important for correct target engagement.

The manuscript is well written and provides certain evidence that dimerization is important for correct target engagement.

The focus lies on the structural validation of the hypothesis that dimerization is important for ADP-ribosylation. Crystal

structures and HDX measurements confirm the dimerization of PARP 15. To fully support the hypothesis that ADP-

ribosyltransferase activity of PARP 15 is regulated by ART domain dimerization, in vivo characterization is needed.

The following suggestions can elucidate the cellular effects of PARP15 dimerization and can strengthen the link between structural findings and the cellular function(s).

1. The authors use a GFP-labeled PARP15 variant. Additional validation of this fusion protein is needed to confirm similar behavior as for the non-GFP-tagged proteins.

2. To provide evidence that dimerization is important for correct target engagement, it would be interesting to identify the targets of ADP-ribosylation and to identify changes in the ADP-ribosylation of target proteins due to the different mutants. Such experiments might also clarify PARP15's role in translation or antiviral defense.

3. While dimerization is shown to be critical for activity in vitro and in cells, the biological contexts where dimerization occurs (e.g., stress granule formation, viral infection) are not experimentally explored. Including cellular stress assays (e.g., oxidative stress, viral mimic treatment) could link dimerization to functional outcomes.

4. The in vitro assays show changes in MARYlation activity. Based on the recombinant expression of PARP15 in E. coli, the enzymes (WT and/or Mutants) might already be entirely ADP-ribosylated. In E. coli, NAD is present in a mM range, and it is expected, based on PARP15's self-ADP-ribosylation activity, that the proteins might already be modified. A possibility to test this is a Western blot using an anti-MAR antibody. If MARYlation can be observed, especially figures such as Fig. 3 need to be reevaluated. Changes in MARYlation of PARP15 might also affect dimerization in vitro.

5. To avoid confusion about in vitro and in vivo experiments, the authors should always emphasize if in vitro conditions were applied, e.g., in Figure 3.

Reviewer #3

(Remarks to the Author)

The authors analyze a relatively under-studied protein, PARP15, largely focusing on the relationship between dimerization and catalysis in the protein, and ultimately proposing an atomic-resolution mechanism. A wide range of techniques including HDX-MS, XL-MS, and x-ray crystallography are employed and data shown in the manuscript includes both in vitro and in cell work. The paper makes a strong and thorough case for the importance of ART domain dimerization in PARP15 function, however some statements are lacking full experimental proof and some revision is required.

Fig. 2D – The HDX-MS experiment was not done in a differential format where monomeric and dimeric PARP15 are compared– rather, the authors infer that reduced D incorporation of the protein near the proposed dimer interface supports their dimerization model. This is a somewhat qualitative assumption. The strongest solution here would be to perform HDX on a protein construct incapable of dimerization, which could be compared to the initial HDX data in what's assumed to be the dimer. At least, the secondary structure of the protein should be shown in the figure to help the reader judge whether the slow D exchange at the proposed interface region is simply a result of a high degree of secondary structure in that area, or a real result of dimerization.

Pg 4, 27-34: Proving interprotein crosslinks between subunits of a homodimer is notoriously difficult. The authors make an intriguing statement that a crosslink detected on multiple overlapping peptides proves unambiguous assignment as in interprotein crosslink. Please elaborate on this to help the reader judge the reliability of the interprotein crosslink, perhaps also including a diagram. The low abundance of intra-subunit crosslinks also makes the existence of the interprotein crosslink questionable, since typically intraprotein crosslinks are found in a much higher abundance than inter- crosslinks. Fig. 2E – It would be easier to visualize the crosslinks in the context of the dimer if the protomers were different colors.

Fig. 3C – Is it possible the increased MARYlation of the dimer is simply a result of the dimeric form bringing substrate in closer proximity, rather than dimerization being required for catalytic activity? Is this interpretation being ruled out by the fluorescent macrodomain overlay in 3D? If so please include one or two sentences summarizing this assay in the text.

Pg. 8, 14-15 – “4 h before fixation, cells were treated with the PARG inhibitor PDD00017273, which we empirically determined enhances PARP15 MARYlation in cells” – if this is a novel finding, some data supporting this should be shown in the supporting information.

Fig. 5C – The D loop should be labeled on this figure. Also, was the ligand observed in the crystal structure? It may also be helpful to show the ligand on the structure as the author's state dimerization has a close relationship with NAD⁺ binding.

Version 1:

Reviewer comments:

Reviewer #1

(Remarks to the Author)

All of my concerns were addressed properly.

Reviewer #2

(Remarks to the Author)

The authors have partially addressed my previous comments and questions.

While substantial work has been carried out in vitro, the conclusions that can be drawn regarding in vivo relevance remain limited. The question of specific targets as well as cellular localization were not addressed by the authors.

It is also somewhat puzzling that PARP15 does not appear to be modified in *E. coli* in the presence of NAD⁺, but only in vitro after addition of NAD⁺ (Supplementary Figure 5). This observation would benefit from further discussion by the authors.

The newly added Figure 7 is a valuable schematic that enhances understanding. However, the lettering is too small, the dotted lines of the circle are difficult to see, and a legend is missing. Addressing these points would improve the clarity of the figure.

Reviewer #3

(Remarks to the Author)

This is a complex study. The authors have addressed my concerns as best they can.

Reviewer #1 (Remarks to the Author)

The paper describes the significance of the dimer formation PARP15, which is a mono-ADP-ribosyltransferase that targets an unknown set of proteins as well as RNA. PARP-15 dimerizes in solution. The author concluded that there is a regulatory mechanism by which dimerization is linked to correct target engagement rather than NAD co-substrate binding and by which the two protomers of the dimer operate independently of one another. PARP15 regulation is significant and interesting as PARP's topic. The dimerization regularization is possible and interesting. However, it is important to identify and reveal the complex structure of PARP15 and its substrate to reveal the regularization. In this meaning, the paper is not enough in Nature Communication.

We thank reviewer #1 for their time and effort.

Major points

The activity was checked by auto-MARylation. Does the author think that there is a similar mechanism between the auto-MARylation and real substrate MARylations? Is it necessary to use real substrate to understand the PARP15 ADP-ribosylation? Finally, I request the figure how the dimerization works for the auto-MARylation and real substrate MARylations. The figure answers my question, including the auto-MARylation description below.

For PARP15 and for most other family members neither the sites nor the purpose of automodification have been identified, nor have the “real” (physiological) substrates been identified and verified.

Automodification has been described *in vitro*, *in cellulo*, and *in situ* for many PARP family members and is considered a physiological activity. One well-studied example is PARP1 which, upon binding to a site of DNA damage, auto-PARylates, regulating its release from chromatin and its enzymatic activity. The effect of automodification on other PARP family members, especially in the MARylating sub-family, is less well studied, but is thought to play a role in their physiological functions. For some PARP enzymes (e.g. PARP10 and PARP14), automodification sites have been identified on the surface far from the active site. Thus, it is commonly accepted that (i) automodification happens in trans, not in cis; and (ii) the automodification we report is not a transient phenomenon in the active site as part of a catalytic mechanism (as frequently observed in other enzyme families, such as acetyltransferases). Considering these aspects, there is no reason to believe that automodification differs mechanistically from trans-modification of other protein targets.

We cannot assess trans-modification activity in lieu of known and confirmed physiological targets for PARP15. However, we observed that our PARP15 Macro2-ART construct (i.e., isoform-2) had much stronger automodification activity than the isolated ART domain. To test whether PARP15 dimerization is necessary for reactions other than automodification, we therefore used the free macrodomain-2 of PARP15 as an alternative “trans-modification” target in our study (see Figure 3f).

Nevertheless, prompted by reviewer #1's question, we have now performed a trans-modification experiment with SRSF protein kinase 2 (SRPK2) as a substrate. Although we are not aware of a study that verified the physiological relevance, SRPK2 was initially identified in a screen for targets of PARP10 (Feijs et al., 2013; doi: 10.1186/1478-811X-11-5), was shown to be modified also by PARP15 (Venkannagari et al., 2013; doi.org/10.1016/j.ejps.2013.02.012) and has been

used in biochemical assays of both enzymes (e.g., Morgan and Cohen, 2015; doi.org/10.1021/acscchembio.5b00213; Chan et al., 2024; doi.org/10.1161/ATVBAHA.124.321522). Notably, our results show that the effect of the dimer interface mutants is similar, if not more pronounced, as in auto-modification reactions (see the new Supplementary Information Figure 5b).

To conclude this part of our response, we wish to remind the reviewer that the various PARP15 mutations we assessed have very similar effects on both auto- and trans-modification activities *in vitro* and in cells.

Finally, the reviewer requests a schematic figure illustrating the binding cycle of PARP15 during auto- and during trans-modification. We have prepared a figure that summarizes the main points of our study and present it as Figure 7 in the new manuscript version. We thank the reviewer for the suggestion as we feel that this makes a strong contribution to clarifying our results.

Auto-MARylation means that the PARP ADP-ribosylates itself.

If dimer exists (AB), does one protomer (A) ADP-ribosylate the same protomer (A)? If dimer exists, one protomer (A) ADP-ribosylates another protomer (B)? Alternatively, AB dimer ADP-ribosylates other dimer (CD)?

This question appears relevant to our analysis of the isolated ART domain under *in vitro* situations, but we remind the reviewer of the domain structure of the two natural PARP15 isoforms, which contain one and two macrodomains, respectively. Macrodomain-2 (present in both isoforms) is the strongly preferred target domain in automodification reactions *in vitro* but assuming a flexible linker between this and the ART domain, automodification might occur either in cis or in trans. Nonetheless, the identification of automodification sites within the ART domain would be necessary to assess whether automodification occurs in cis or trans. Then, the scenarios listed by the reviewer could be tested by mutagenesis in combination with a method that separates cis- from trans-automodification of the PARP15 ART dimer.

I would like to know what does the author think about auto-MARylation.

The physiological consequences of PARP15 auto-MARylation are not clear. We have addressed a question posed by reviewer #2 by experiment and we now know that PARP15 automodification does not affect its propensity to dimerize (please see below, as well as the new Fig. 3g). It is conceivable that automodification of specific sites might regulate PARP15 interaction with the targets of its three domains differentially, perhaps by blocking putative protein – protein interaction sites. Given the links between MARylation and the ubiquitin-proteasome system, automodification might also act as a “timer” that ensures PARP15 degradation following a period of activity. Given what we describe above, an understanding of these processes will require the identification of automodification target sites and their recognition by other enzymes including the ADP-ribosyl glycohydrolases.

The author concluded that there is a regulatory mechanism by which dimerization is linked to correct target engagement rather than NAD co-substrate binding and by which the two protomers of the dimer operate independently of one another.

>

How does the author conclude as "dimerization is linked to correct target engagement"?

We thank the reviewer for spotting this misleading phrasing in the abstract. We have now exchanged “is linked to” for “enables”, which is the correct conclusion of our HDX-MS analysis of

loop movements in the PARP15 dimer. The conclusion is also illustrated in our new schematic figure 7.

Minor piints

we treated proteins with BS3, a chemical crosslinker with a spacer arm length of 11.4 Å

→ Please add what reaction occurs using BS3.

We now added the reaction specificity of BS3 in the text and provided additional references.

Fig2E

Enlarge E (Amino acids). They are too small. What is the inter-protomer peptide in yellow?

The purple line shows the inner-protomer peptide. Add the explanation in the legend.

We now enlarged the amino acid sequence in Figure 2e and extended the legend according to the reviewer's suggestions.

Moreover, in this figure, it is better to explain the mutational residue used in this paper.

R576, D665, and H559. Add the explanation of the role of these residues based on the PARP structure as a supplemental figure.

We do not see the need to add results concerning these mutations in Figure 2, and this would disrupt the logic of our manuscript. Instead, we first establish the biophysical observations that indicate dimers, and then present Figure 3a, where the interface mutations are explained and depicted in a schematic fashion.

The H559Y mutation affects the catalytic site, not the interface. This mutation was not designed in this study, and we refer to the original article in which it is described in detail.

Fig.3CD

C: This shows auto-MARylation of PARP15.

We now state this specifically in the figure legend.

D: What is a fluorescent macrodomain overlay assay?

We provide reference to the original article that describes this method to detect and quantify target MARylation using NAD⁺ (as opposed to an NAD⁺ analog, such as N6-biotin-NAD⁺, needed for detection) and a fluorescent macrodomain as sensor.

Fig3F

I understand the assay macrodomain-2 as substrate. However, I do not understand the meaning why the author used H559Y. Please insert the graphic figure as a supplemental figure.

(p19 L9~L20:

When PARP15 macrodomain2 was used as a substrate~0.2mM TCEP.

I do not understand “ was mixed with titrations of PARP15 ART H559Y”.)

These two questions address one of the key experiments in our study. In this experiment we added the inactive H559Y mutant to raise the concentration of ART domain above the approximate K_d for dimerization (originally established by the experiment in Fig. 2e), without adding additional catalytically active units. We have now prepared a detailed schematic figure that outlines the setup of this experiment (Supplementary Figure 5d) and described the assay in more detail in the main text.

Fig.4

Four hours post-transfection, cells were treated with Saruparib (AZD5305) to diminish PARP1 contribution to ADP-ribosylation levels, and 4 h before fixation, cells were treated with the PARG inhibitor PDD00017273, which we empirically determined enhances PARP15 MARYlation in cells.

>

Saruparib inhibits PARP1. Does Saruparib not inhibit other PARPs, including PARP5?

AZD5305 (Saruparib) is presently considered the most selective PARP1 inhibitor, and the authors did not detect an effect on PARP15 (Johannes et al., 2021; DOI: 10.1021/acs.jmedchem.1c01012).

What is the mechanism to enhance PARP15 using PARG inhibitor?

The regulation of PARP15-dependent MARYlation by poly(ADP-ribose) glycohydrolase (PARG) has been discovered by our collaborator and co-author Michael S. Cohen. It is the topic of a forthcoming publication by Sanderson, Cohen et al. We anticipate that we will be able to refer to the published paper at a later stage. Here, we show experimental evidence supporting a need for a PARG inhibitor to stabilize PARP15-derived MARYlation in cells, kindly provided by Sanderson & Cohen. The figure below shows Western blots of lysates from HEK293T cells overexpressing PARP15, which had been treated with siRNA to knock down each of six ADP-ribosyl glycohydrolases (PARG; TARG/C6ORF130; ARH1; ARH3; MacroD1; or MacroD2). Cells were treated with either PARG inhibitor PDD or DMSO 4 hours before harvest. Together, these results suggest that absence of PARG activity results in stable PARP15 derived MARYlation signal. The mechanism of this stability will be reported in the Sanderson et al. paper.

Fig.5CD

Show the D-loop in the figure. (Base of the D-loop?)

We now colored the D-loop in the R576E mutant structure differently and labeled it in Figure 5c. In addition, we now provide an annotated view of the PARP15 ART dimer that details these structural features (Supplementary Fig. 9).

The “base of the D-loop” is called (by us and others) the N-terminus of that loop. Although we stated, “N-terminal base...” in the text, we have now changed the description in the figure legend to “N-terminus of the D-loop”.

P10 l24

We hypothesized ~>

Add the explanation of NAD binding cooperativity.

If NAD binds one of the protomers, the subsequent NAD binding is easy to bind to the other monomer in the dimer. The author analyzed cooperativity from the bottom figure. N is the cooperativity from the hill plot? Please add the explanation.

We now added the definition of binding cooperativity in the text. "n" is the stoichiometry of the binding reaction and is one of the parameters extracted from the fit of a one-site binding model to the data. As such, n is part of the model fit.

Why does the ART domain dimer work better for the substrate? Please explain a possible mechanism. Again, it is better to depict in the final figure (How do the dimer (mac1-mac2-ART) and substrate form the complex and express the activity). These figures could explain the D-loop interaction of the dimer and the opening for the NAD binding site, as the author suggested (p14 l8 and p14 l11~l27).

We now added a new Figure 7, illustrating our current working model. It explains how ART domain dimerization alters loop conformations and flexibility, leading to the activation of catalysis. We also added Supplementary Figure 9, which depicts these loops in the PARP15 ART crystal structure.

For the reasons stated above, a schematic image showing the full-length PARP15 in complex with a protein substrate would be purely speculative. As this content is outside of the scope of our data, we refrain from showing this type of image.

P15 l8~l9

Four ADP-ribose binding domains> The dimer includes two ART domains.

Isoform 1 of PARP15 has two ADP-ribose binding domains (i.e. macrodomains), which we discuss as well as illustrate in our manuscript. The ADP-ribosyltransferase domain (ART domain) is not an ADP-ribose binding domain, as ADP-ribose is the product of the transferase reaction. Thus, the dimer of PARP15 isoform-1 contains four macrodomains and two ART domains.

I expect future studies of the complex structure PARP15 and the substrate.

We agree that this is an important future goal – for all PARP family members. Although we and others have pursued the identification of the physiological substrates (targets) for PARP15, they remain unknown and putative targets remain unverified. Furthermore, a complex structure together with target has never been clarified for *any* member in the PARP family - including the heavily studied clinical target PARP1, the first crystal structure of which has been determined as early as 1996 (A. Ruf and G. Schulz, PNAS) and tankyrase, the domain structure and regulation of which has been documented by experimental methods (L. Mariotti et al., Nature 2022). Unfortunately, the referee’s request to add a PARP15 target complex structure to this study is not viable at present.

Reviewer #2 (Remarks to the Author)

The manuscript "Regulation of ADP-ribosyltransferase activity by ART domain dimerization in PARP15" submitted by Ebenwaldner et al. provides valuable insights into PARP15's protein structure, thereby aiming to elucidate catalytic mechanisms *in vitro* and *in vivo*.

The authors apply various structural biology technologies such as crystallography and HDX-MS to verify the dimerization of PARP15. Additionally, they characterize the influence of dimerization on the catalytic properties of PARP15, mostly *in vitro*. The data provide evidence that dimerization is important for correct target engagement.

The manuscript is well written and provides certain evidence that dimerization is important for correct target engagement.

The focus lies on the structural validation of the hypothesis that dimerization is important for ADP-ribosylation. Crystal structures and HDX measurements confirm the dimerization of PARP 15. To fully support the hypothesis that ADP-ribosyltransferase activity of PARP 15 is regulated by ART domain dimerization, *in vivo* characterization is needed.

The following suggestions can elucidate the cellular effects of PARP15 dimerization and can strengthen the link between structural findings and the cellular function(s).

We thank reviewer #2 for their time and effort.

1. The authors use a GFP-labeled PARP15 variant. Additional validation of this fusion protein is needed to confirm similar behavior as for the non-GFP-tagged proteins.

In the cellular experiments, we used mutations the effects of which we had characterized in the preceding biochemistry experiments. Our experiments containing the GFP-tagged wild type and these mutant enzymes provided the controls the reviewer asks for: Our results showed that each variant indeed does display similar behavior as the untagged versions; i.e., the activity observed for each variant *in vitro* is replicated in the GFP-tagged version of that variant in cells. Thus, this confirms that GFP-tagging does not significantly influence the behavior of PARP15.

2. To provide evidence that dimerization is important for correct target engagement, it would be interesting to identify the targets of ADP-ribosylation and to identify changes in the ADP-ribosylation of target proteins due to the different mutants. Such experiments might also clarify PARP15's role in translation or antiviral defense.

We agree with Reviewer #2 that identifying PARP15 targets would be of great interest as a next step toward understanding PARP15 physiological functions. PARP target identification has been subject to the field for more than a decade, but is difficult to tackle: the vast majority of putative targets for PARP10 and PARP14 (the best studied MARYlating enzymes) have yet to be verified.

Our manuscript describes PARP15 dimerization as a novel and thus far unique requirement for catalytic activity among PARP enzymes. PARP15 target identification and verification, requiring a specialized set of methods and expertise, would constitute a highly significant discovery of its own standing. We fully agree that our study provides insights and research tools (in the form of dimerization incompetent mutants) for studies into the physiological roles of PARP15.

3. While dimerization is shown to be critical for activity *in vitro* and in cells, the biological contexts where dimerization occurs (e.g., stress granule formation, viral infection) are not experimentally explored. Including cellular stress assays (e.g., oxidative stress, viral mimic treatment) could link dimerization to functional outcomes.

As with the reviewer's previous comment, we fully agree that these are important questions to address in order to understand how PARP15 works. We contribute important knowledge regarding the most basic functions of PARP15 – knowledge that compromises the interpretation of some of the work that has been done during the past 15 years. In the future, researchers will be able to incorporate our findings into their experimental design, e.g. by considering the K_d of dimer formation and by making use of our dimer disrupting mutations. Our main expertise lies within biochemistry, biophysics, and structural biology – disciplines that are insufficient to explore the questions suggested by the reviewer.

4. The in vitro assays show changes in MARYlation activity. Based on the recombinant expression of PARP15 in *E. coli*, the enzymes (WT and/or Mutants) might already be entirely ADP-ribosylated. In *E. coli*, NAD is present in a mM range, and it is expected, based on PARP15's self-ADP-ribosylation activity, that the proteins might already be modified. A possibility to test this is a Western blot using an anti-MAR antibody. If MARYlation can be observed, especially figures such as Fig. 3 need to be reevaluated. Changes in MARYlation of PARP15 might also affect dimerization in vitro.

This is an important aspect of PARP biochemistry, especially as long as we do not understand the functions of PARP automodification. As part of routine quality assertion, we have previously monitored automodification of recombinant PARP enzymes during expression in *E. coli* and found only low levels of automodification in the most active PARP enzymes (such as PARP1 and PARP10; unpublished findings). This is puzzling to us, but a discussion of the possible mechanisms behind this observation would be out of place here.

Nevertheless, we fully agree with reviewer #2 that we must address here specifically whether our PARP15 expression constructs are obtained as pre-modified during expression in *E. coli*. As suggested, we used an anti-MAR specific antibody to test ADP-ribosylation on PARP15 ART domains (wild-type, catalytically inactive H559Y mutant, and interface mutant R576E). To compare levels of potential modification with the levels of automodification typically observed under our reaction conditions, we also added NAD^+ to aliquots of each of them and probed auto-MARYlation over a time course of 30 minutes. The Western blot in our new Supplementary Figure 5a shows that neither of these three constructs is pre-modified to detectable levels in *E. coli*. Upon incubation with $50 \mu M$ NAD^+ for 10 minutes, we can detect ADP-ribosylation on the PARP15 wild-type ART domain, but not on the R576E mutant. The H559Y mutant shows minimal levels of auto-modification, indicating residual catalytic activity.

Regarding the final comment of this point by reviewer #2; we agree it was important to test whether auto-MARYlation on PARP15 would interfere with its dimerization. Therefore, we auto-MARYlated PARP15 and tested it alongside un-modified PARP15 in a sedimentation velocity experiment (Analytical Ultracentrifugation). The results show that automodification does not affect dimerization of PARP15. These results are shown as a new panel in Figure 3g. A Western blot experiment performed on the samples recovered from AUC cells after the run had completed confirms that PARP15 was indeed effectively automodified in the NAD^+ conditions (Supplementary Figure 5c).

Having thus established AUC conditions for PARP15, we went on to determine the K_d of dimerization (although this was not requested by any of the reviewers). Under NAD^+ -free conditions our data suggest a K_d of 313 nM (new Fig. 1j,k, new Supplementary Fig. 1, and new Supplementary Table 2), which seems consistent with our argument based on the activity assays

in Fig. 3e that the K_d for dimer formation must lie in the mid-nanomolar range. We have added brief descriptions of these results in the appropriate text sections.

5. To avoid confusion about in vitro and in vivo experiments, the authors should always emphasize if in vitro conditions were applied, e.g., in Figure 3.

We thank the reviewer for this reminder; we have made text changes accordingly.

Reviewer #3 (Remarks to the Author):

The authors analyze a relatively under-studied protein, PARP15, largely focusing on the relationship between dimerization and catalysis in the protein, and ultimately proposing an atomic-resolution mechanism. A wide range of techniques including HDX-MS, XL-MS, and x-ray crystallography are employed and data shown in the manuscript includes both in vitro and in cell work. The paper makes a strong and thorough case for the importance of ART domain dimerization in PARP15 function, however some statements are lacking full experimental proof and some revision is required.

We thank reviewer #3 for their time and effort.

Fig. 2D – The HDX-MS experiment was not done in a differential format where monomeric and dimeric PARP15 are compared– rather, the authors infer that reduced D incorporation of the protein near the proposed dimer interface supports their dimerization model. This is a somewhat qualitative assumption. The strongest solution here would be to perform HDX on a protein construct incapable of dimerization, which could be compared to the initial HDX data in what's assumed to be the dimer. At least, the secondary structure of the protein should be shown in the figure to help the reader judge whether the slow D exchange at the proposed interface region is simply a result of a high degree of secondary structure in that area, or a real result of dimerization.

The proposed HDX-MS experiment (comparison of monomeric and dimeric PARP15) has indeed been performed and is presented in Figure 6. Figure 2 establishes that PARP15 dimerizes via its ART domain, and that conclusion is drawn from HDX-MS results together with other data. For the sake of clarity, we chose to keep the general structure of the manuscript unchanged.

We do agree with the reviewer that slow deuterium exchange correlates with particularly stable secondary structure elements and that it is impossible to tell whether this is the case from Figure 2d alone. Thus, we refer reviewer #3 to Supplementary Fig. 2d, where the relevant data is represented on the cartoon model of the m2-ART construct at all tested time points of the experiment. From this figure, it is apparent that the deuterium exchange at the interface on the ART domain is slowed down compared to other surface-exposed areas on macrodomain 2 or the ART domain. For clarity, we now added more labels to panel d in Supplementary Fig. 2 and believe that this will resolve the reviewer's concern.

Pg 4, 27-34: Proving interprotein crosslinks between subunits of a homodimer is notoriously

difficult. The authors make an intriguing statement that a crosslink detected on multiple overlapping peptides proves unambiguous assignment as in interprotein crosslink. Please elaborate on this to help the reader judge the reliability of the interprotein crosslink, perhaps also including a diagram. The low abundance of intra-subunit crosslinks also makes the existence of the interprotein crosslink questionable, since typically intraprotein crosslinks are found in a much higher abundance than inter- crosslinks.

We thank the reviewer for pointing out the necessity to better describe our interpretation of our results. Although the sequences of the crosslinked peptides were shown along their mass spectra in the original manuscript version, we have now added an additional panel to Supplemental Fig. 3 and improved our description in the text.

Regarding the reviewer's second point, it is important to remember that the number of "detected crosslinks" is always a matter of thresholds, i.e. a matter of interpretation. Here, we did indeed detect a higher abundance of masses indicating intra-, but also inter-protomer crosslinks. However, the majority of those are not listed here, as we applied stringent filtration criteria throughout the analyses in order to avoid false positives. The score cut-off was set particularly high; only cross-links that were detected more than once were included; and only those with trustworthy mass spectra were listed in our figures and in Supplementary Table 4, with one example of mass spectra for each included in Supplementary Figure 3. We believe that this is the proper way to proceed. Readers with a special interest will find the full list of masses obtained from our experiments in the PRIDE repository.

Fig. 2E – It would be easier to visualize the crosslinks in the context of the dimer if the protomers were different colors.

We thank reviewer #3 for this suggestion. We have updated panel 2e in this manuscript version. In addition, we have added another schematic to Supplementary Fig. 3. We are confident that with these changes a reader will be able to distinguish the two protomers and the positions of individual crosslinks.

Fig. 3C – Is it possible the increased MARylation of the dimer is simply a result of the dimeric form bringing substrate in closer proximity, rather than dimerization being required for catalytic activity? Is this interpretation being ruled out by the fluorescent macrodomain overlay in 3D? If so please include one or two sentences summarizing this assay in the text.

In the scenario proposed by the reviewer, automodification would occur only within the dimer, from each protomer to the other. While we cannot exclude this scenario, it appears quite unlikely, as all signals in our automodification assays would have to derive from those few potential target residues able to reach into the active site of the other protomer within the dimer.

However, as a mechanism for PARP15 activation, the reviewer's scenario is ruled out by the results of our trans-modification assays, namely, the macrodomain-2 modification experiment in Fig. 3f, the new SRPK2 modification experiment in Supplementary Fig. 5b, as well as the cellular experiments in Supplementary Fig. 6, where the dimer interface mutations have the same abrogating effects on catalytic activity as in the automodification experiments.

An illustration of a possible mechanism for catalytic activation is provided in the new schematic figure 7, and we have added a brief description in the Discussion section.

Pg. 8, 14-15 – “4 h before fixation, cells were treated with the PARG inhibitor PDD00017273, which we empirically determined enhances PARP15 MARYlation in cells” – if this is a novel finding, some data supporting this should be shown in the supporting information.

We provide evidence for the need of these experimental conditions above. Please see our response to referee #1 query.

Fig. 5C – The D loop should be labeled on this figure. Also, was the ligand observed in the crystal structure? It may also be helpful to show the ligand on the structure as the author’s state dimerization has a close relationship with NAD⁺ binding.

We thank the reviewer for this suggestion. We have labelled the D-loop in the new Figure 5c.

The ligand was indeed observed in the electron density during X-ray crystallography. We now have chosen a more distinct color for the ligand to improve clarity, and have pointed it out in the new figure legend and in the main text.

The authors have partially addressed my previous comments and questions. While substantial work has been carried out in vitro, the conclusions that can be drawn regarding in vivo relevance remain limited.

The topic of our study is the molecular detail of a newly discovered phenomenon that will change the way we study and understand PARP15 and likely other PARP enzymes: our results open for a new layer of regulation of MARylation activity by interference with ART domain dimerization through protein binding, PTM and ART domain local concentration. Considering these important implications, and the width of our experimental work, we believe it will be sufficient at this stage to provide strong evidence for in-cellulo relevance of the phenomenon. In a broader perspective, we agree in full with the referee that our findings must now be translated to study the functional details of PARP15 in cells.

The question of specific targets as well as cellular localization were not addressed by the authors.

The pursuit of PARP targets is carried out by a handful of highly specialized labs. We do not believe it is meaningful to include our own attempts at such analyses in this study.

None of the referees asked us to further address PARP15 cellular localization in their comments.

It is also somewhat puzzling that PARP15 does not appear to be modified in *E. coli* in the presence of NAD⁺, but only in vitro after addition of NAD⁺ (Supplementary Figure 5). This observation would benefit from further discussion by the authors.

We have demonstrated, using the method requested by Referee #2, that the PARP15 protein we study is not pre-modified when entering our analyses - an important quality control.

Furthermore, we have demonstrated with our AUC experiments that even if PARP15 were pre-modified, this would not influence its dimerization (as the referee had suggested it might). With that, we expect this new data will raise readers' confidence in our experimental results and we thank the referee for the suggestion.

On the other hand, the fate of PARP auto-MARylation activity during recombinant production, interesting or not, is related neither to the topic of our study nor to PARP15 specifically and has thus no legitimate place in our discussion section.

The newly added Figure 7 is a valuable schematic that enhances understanding. However, the lettering is too small, the dotted lines of the circle are difficult to see, and a legend is missing. Addressing these points would improve the clarity of the figure.

We thank the referee for noting this. We have increased the line weight; the lettering was prepared according to the figure guidelines and we are thus confident that the lettering will be legible at final print size. We have also expanded the figure description in the legend.